# Selective Inhibition of Deamidated Triosephosphate Isomerase by Disulfiram, Curcumin, and Sodium Dichloroacetate: Synergistic Therapeutic Strategies for T-Cell Acute Lymphoblastic Leukemia in Jurkat Cells

**DOI:** 10.3390/biom14101295

**Published:** 2024-10-13

**Authors:** Luis A. Flores-López, Ignacio De la Mora-De la Mora, Claudia M. Malagón-Reyes, Itzhel García-Torres, Yoalli Martínez-Pérez, Gabriela López-Herrera, Gloria Hernández-Alcántara, Gloria León-Avila, Gabriel López-Velázquez, Alberto Olaya-Vargas, Saúl Gómez-Manzo, Sergio Enríquez-Flores

**Affiliations:** 1Laboratorio de Biomoléculas y Salud Infantil, CONAHCYT-Instituto Nacional de Pediatría, Mexico City 04530, Mexico; 2Laboratorio de Biomoléculas y Salud Infantil, Instituto Nacional de Pediatría, Mexico City 04530, Mexico; ignaciodelamora@ciencias.unam.mx (I.D.l.M.-D.l.M.); garciaitzhel@gmail.com (I.G.-T.); glv_1999@ciencias.unam.mx (G.L.-V.); 3Posgrado en Ciencias Biológicas, (Maestría), Universidad Nacional Autónoma de México, Mexico City 04510, Mexico; cmmalagonr@gmail.com; 4Instituto Tecnológico y de Estudios Superiores de Monterrey, Campus Ciudad de México, Mexico City 14380, Mexico; yoalli.martinez@tec.mx; 5Laboratorio de Inmunodeficiencias, Instituto Nacional de Pediatría, Mexico City 04530, Mexico; lohegabyqbp@gmail.com; 6Departamento de Bioquímica, Facultad de Medicina, Universidad Nacional Autónoma de México, Apartado Postal 70-159, Mexico City 04510, Mexico; ghernandez@bq.unam.mx; 7Departamento de Zoología, Escuela Nacional de Ciencias Biológicas, Instituto Politécnico Nacional, Carpio y Plan de Ayala S/N, Casco de Santo Tomás, Ciudad de México 11340, Mexico; leonavila60@yahoo.com.mx; 8Trasplante de Células Madre y Terapia Celular, Instituto Nacional de Pediatría, Mexico City 04530, Mexico; alberto.olaya@yahoo.com.mx; 9Laboratorio de Bioquímica Genética, Instituto Nacional de Pediatría, Mexico City 04530, Mexico; saulmanzo@ciencias.unam.mx

**Keywords:** cancer, leukemia, glycolysis, TPI, repurposing drugs

## Abstract

T-cell acute lymphoblastic leukemia (T-ALL) is a challenging childhood cancer to treat, with limited therapeutic options and high relapse rates. This study explores deamidated triosephosphate isomerase (dTPI) as a novel therapeutic target. We hypothesized that selectively inhibiting dTPI could reduce T-ALL cell viability without affecting normal T lymphocytes. Computational modeling and recombinant enzyme assays revealed that disulfiram (DS) and curcumin (CU) selectively bind and inhibit dTPI activity without affecting the non-deamidated enzyme. At the cellular level, treatment with DS and CU significantly reduced Jurkat T-ALL cell viability and endogenous TPI enzymatic activity, with no effect on normal T lymphocytes, whereas the combination of sodium dichloroacetate (DCA) with DS or CU showed synergistic effects. Furthermore, we demonstrated that dTPI was present and accumulated only in Jurkat cells, confirming our hypothesis. Finally, flow cytometry confirmed apoptosis in Jurkat cells after treatment with DS and CU or their combination with DCA. These findings strongly suggest that targeting dTPI represents a promising and selective target for T-ALL therapy.

## 1. Introduction

Acute lymphoblastic leukemia (ALL) is a hematological malignancy characterized by the rapid and uncontrolled proliferation of immature lymphoblasts within the bone marrow and peripheral blood. It constitutes the most prevalent childhood cancer, with B-cell precursors (B-ALL) accounting for approximately 80% of cases, while T-cell precursors (T-ALL) comprise 12–15% [1]. Historically, T-ALL has been associated with a poorer prognosis compared to B-ALL; however, recent advancements in therapeutic protocols have significantly improved survival rates in pediatric T-ALL [2]. Despite these improvements, achieving complete remission still requires highly intensive treatment regimens, and managing relapsed T-ALL remains challenging due to the limited therapeutic options and the development of resistance to standard treatments [3].

Although significant progress has been made in ALL treatments, including chemotherapy, targeted therapies, and stem cell transplantation, relapse rates remain high, especially for certain patient groups, and treatment resistance continues to be a substantial challenge. The five-year survival rate for pediatric patients has improved by over 85%, but for adults, it remains below 40% [4]. These statistics underline the critical need for novel therapeutic strategies and the identification of innovative therapeutic targets and agents.

A promising avenue of investigation involves targeting the metabolic reprogramming observed in cancer cells, particularly the upregulation of aerobic glycolysis to satisfy their elevated bioenergetic demands [5]. In ALL, as in many other cancers, glucose uptake is markedly elevated. Additionally, the hypoxic microenvironment surrounding hematopoietic stem cells further promotes glycolysis upregulation [6]. This metabolic reprogramming not only provides cancer cells with the energy necessary for rapid proliferation but also supplies critical biosynthetic precursors that support their growth and survival.

Within this context, in recent years, there has been increasing interest in triosephosphate isomerase (TPI) due to its potential role in cancer metabolism and tumor progression. TPI catalyzes the reversible interconversion of dihydroxyacetone phosphate (DHAP) and glyceraldehyde-3-phosphate (G3P), playing a central role in glycolysis by generating a greater abundance of G3P molecules, which are subsequently funneled towards the production of ATP and NADH, among others. Given its centrality within the glycolytic cascade, TPI has emerged as a potential target for therapeutic intervention [7]. Notably, post-translational modifications (PTM), such as deamidation, have been identified, which can significantly impact the activity and stability of enzymes, including TPI [8]. Deamidated TPI (dTPI) exhibits distinct biochemical properties compared to its non-deamidated counterpart (n-dTPI), including increased permeability to hydrophobic compounds, which may enhance its susceptibility to therapeutic agents [9].

Drug repurposing offers a strategic advantage by leveraging established safety profiles of existing drugs to expedite the development of novel anti-cancer therapeutics [10]. Tetraethylthiuram disulfide, known as disulfiram (DS), is a hydrophobic, symmetrical dithiocarbamate with thiol-reactive properties currently used to treat alcoholism [11]. DS has demonstrated promising anti-cancer activity, including its ability to target proteins such as MD-2, a critical cofactor of Toll-like receptor 4, potentially suppressing pro-inflammatory signaling cascades [12]. Additionally, it inhibits AKT phosphorylation, thereby disrupting cell survival pathways crucial for tumor maintenance [13]. It also induces the inhibition of nuclear protein localization protein 4 (NPL4), impairing protein degradation within the ubiquitin-proteasome system, which compromises cancer cell viability [14], among others.

Curcumin (CU), a hydrophobic naturally occurring polyphenol derived from the Asian plant *Curcuma longa*, exhibits potent anti-cancer activity by modulating multiple signaling pathways and interacting with a variety of cellular targets [15].

Molecular docking simulations offer valuable insights into potential binding affinities between small molecules and protein targets. For instance, these simulations reveal that both DS and CU exhibit strong binding affinity to hydrophobic pockets within proteins. DS, for example, effectively binds to the hydrophobic TIR domain of TLR4 [16], a pivotal activator of the innate immune response and a critical player in combating bacterial infections. Similarly, CU shows high affinity for hydrophobic cavities in phosphoinositide 3-kinase (PI3K) [17], a key player in the PI3K/AKT/mTOR signaling pathway, which controls cell proliferation. The propensity to bind in hydrophobic cavities in proteins by DS and CU warrants further investigation to determine their potential interaction with dTPI.

In addition to DS and CU, sodium dichloroacetate (DCA), a small-molecule inhibitor of pyruvate dehydrogenase kinase (PDK), has shown promise in cancer therapy by modifying tumor metabolism. While not yet an approved drug, DCA has been used for over 50 years in the treatment of illnesses such as diabetes mellitus, lipid and lipoprotein disorders, and congenital lactic acidosis, among others [18].

DCA’s mechanism involves the inhibition of PDK, an important regulator of the pyruvate dehydrogenase complex (PDC), which governs the entry of pyruvate into the mitochondrial tricarboxylic acid (TCA) cycle [19]. By inhibiting PDK, DCA promotes the conversion of pyruvate to acetyl-CoA, potentially forcing a metabolic shift from aerobic glycolysis towards oxidative phosphorylation. This disruption of metabolic flexibility in cancer cells can lead to decreased proliferation and increased apoptosis. Therefore, DCA is a promising strategy to sensitize T-ALL cells by targeting their altered metabolic state. Combining DCA with other therapeutic agents such as DS or CU could enhance treatment efficacy by sensitizing cancer cells to subsequent therapies.

Our study aimed to explore the therapeutic potential of dTPI in a cell model of T-ALL (Jurkat cells). Specifically, we assessed the ability of DS and CU to bind to and inhibit dTPI activity. Molecular docking simulations revealed a higher binding affinity of both compounds to dTPI compared to its non-deamidated counterpart (n-dTPI). Subsequent experimental assays confirmed that DS and CU completely inhibited recombinant dTPI enzymatic activity, with minimal effects on recombinant n-dTPI activity.

In Jurkat cells (a model of human T-cell leukemia), treatment with DS and CU resulted in reduced cell viability and TPI cellular activity, with IC_50_ values for cell viability of 454 µM and 829 µM, respectively. Notably, pre-treatment with DCA followed by combination treatment with DS or CU demonstrated notable synergistic effects. DCA, likely sensitized cancer cells, enabling reduced concentrations of DS or CU to effectively inhibit both cell viability and TPI cellular activity. These treatments had minimal effects on normal T lymphocytes.

Importantly, Jurkat cells exhibited an accumulation of acidic isoforms of TPI, corresponding to deamidated enzyme, which were absent in normal T lymphocytes. This accumulation increased further pre-treatment with DCA, followed by combination treatment with DS or CU. TPI inactivation by these compounds also led to elevated levels of methylglyoxal (MGO) and advanced glycation end products (AGEs) within Jurkat cells. Since MGO and AGEs are known to induce apoptosis, flow cytometry assays confirmed that cell death predominantly occurred through the apoptotic pathway, with minimal effects on normal T lymphocytes.

In summary, our findings demonstrate the potential of dTPI as a therapeutic target in Jurkat cells. DS and CU emerge as promising candidates due to their ability to selectively ablate endogenous TPI enzymatic activity and viability of T-ALL cells. Furthermore, the combined treatment with DCA markedly enhanced the anticancer effects in Jurkat cells while largely preserving the integrity of normal T lymphocytes. The synergistic interaction between DCA-DS or DCA-CU strongly suggests that DCA enhances the effectiveness of the compounds on Jurkat cells, thereby heightening their sensitivity to TPI inhibition and promoting apoptotic cell death.

## 2. Materials and Methods

### 2.1. Reagents and General Materials

The following list corresponds to the reagents and materials used in the experimental assays. Other reagents mentioned in the manuscript were procured from Sigma-Aldrich (St. Louis, MO, USA). Luria-Bertani (LB) medium and isopropyl-β-D-thiogalactopyranoside (IPTG) were provided by VWR Life Science Products (Pensilvania, PA, USA). Glycerol-3-phosphate dehydrogenase (α-GDH) and reduced nicotinamide adenine dinucleotide (NADH) were sourced from Roche (Mannheim, Germany). Immobilized metal affinity chromatography (IMAC) resin was obtained from Bio-Rad (Hercules, CA, USA). Sephadex G-25 fine resin was purchased from Amersham Biosciences (Amersham, UK), while Amicon Ultra 30 kDa filters were from Merck-Millipore Corporation (Billerica, MA, USA). Fetal bovine serum (FBS), penicillin, streptomycin, and trypsin EDTA solutions were obtained from Invitrogen (Carlsbad, CA, USA).

### 2.2. Molecular Docking of Non-Deamidated TPI and Deamidated TPI with Disulfiram and Curcumin

Molecular docking assays were performed using the crystallographic coordinates of human non-deamidated TPI (n-dTPI) and human deamidated TPI (dTPI), downloaded from the Protein Data Bank (PDB), with PDB codes: 2jk2 and 4unk, respectively [9,20]. Solvents, ions, and ligands were removed using PyMOL version 2.5.0 (Schrödinger Inc., New York NY, USA). Subsequently, the structures were energetically minimized with UCSF Chimera software (1.18, UCSF, San Francisco, CA, USA) [21], and the resulting new coordinates were used for docking calculations.

Biologically relevant ligands, DS and CU, both potential inhibitors of dTPI activity, were utilized in this study. Their molecular structures were obtained from the ZINC database (https://zinc.docking.org accessed on 10 July 2023) and energy minimized using Avogadro version 1.2. Protein structures were prepared by adding hydrogen atoms and Kollman charges (6.00 and 3.999, respectively) using AutoDock Tools (ADT) version 1.5.6 [22]. Molecular docking was conducted utilizing the “CB-Dock” server [23], employing the default configuration. The resulting output files were saved in .pdb format. Subsequently, the interactions between the compounds and the interface of both enzymes were analyzed and visualized using the PyMOL Molecular Graphics System software (version 2.5.0, Schrödinger, LLC, New York, NY, USA).

### 2.3. Cloning, Overexpression, and Purification of Human TPI Recombinant Enzymes

We employed previously cloned genes encoding human TPI with His-tag and recognition site for Tobacco Etch Virus Protease (TEVp). These genes and constructs facilitated the purification of non-deamidated TPI (n-dTPI) as well as its variants: single-deamidated (dTPI) and double-deamidated (ddTPI). Therefore, our study encompassed the recombinant TPI wild-type enzyme without mutations (WT) along with mutated TPI enzymes designed to mimic the corresponding single and double deamidations, specifically N16D and N16D/N72D, respectively [9]. The plasmids containing the inserts *pET3a-HisTEV-hstpi-wt*, *pET3a-HisTEV-hstpi-n16d*, and *pET3a-HisTEV-hstpi-n16d/n72d* were employed to transform *Escherichia coli* strain BL21-CodonPlus-RIL. The overexpression and purification of the recombinant TPIs were conducted following established protocols [9] as follows: Purified TPIs were concentrated using Centricon filters with a 30 kDa cutoff for WT (n-dTPI) and a 10 kDa cutoff for N16D (dTPI) and N16D/N72D (ddTPI). Upon reaching a volume of 0.5 mL, this process was repeated three times following the addition of 5 mL of TEA buffer (100 mM triethanolamine, pH 7.4). To remove the His-TEV tag, the protein suspension was incubated with TEVp (1:50, TEVp/TPI) and 1 mM dithiothreitol (DTT) for 16 h at room temperature. Subsequently, the samples were loaded onto an IMAC resin column previously equilibrated with TEA buffer. The tag-free enzymes were recovered, concentrated, precipitated with 75% ammonium sulfate, and stored at 4 °C until use.

Prior to assays, recombinant proteins were equilibrated in TEA buffer and incubated at 4 °C in the presence of 5 mM DTT for 1 h. To remove the reducing agent, the samples were filtered by centrifugation on a 1 mL column loaded with Sephadex G-25 fine resin previously equilibrated with TEA, and the protein concentration was estimated by absorbance at 280 nm using an extinction coefficient of ε  =  33,460 M^−1^ cm^−1^ [24]. Protein purity and integrity were confirmed by sodium dodecyl sulfate-polyacrylamide gel electrophoresis (16% SDS-PAGE) and colloidal Coomassie brilliant blue staining.

### 2.4. Enzyme Activity Assays of n-dTPI and dTPI VS DS and CU

Stock solutions of 100 mM DS and CU were prepared in 100% ethanol and further diluted with TEA buffer to a final concentration of 5 mM. For enzyme activity assays in the presence of compounds, recombinant proteins were incubated at 0.2 mg/mL in the presence of DS or CU at concentrations of 0, 50, 100, 250, 500, and 1000 µM for DS and 0, 100, 500, 750, 1000, and 1500 µM for CU during 2 h at 37 °C. The control condition included the protein without the compounds but with 1.5% ethanol, equivalent to the highest percentage used. Subsequently, aliquots were taken and diluted at concentrations of 5 ng/mL and 50 ng/mL for the n-dTPI and dTPI, respectively.

Enzyme activity was spectrophotometrically measured using a Cary 50 spectrophotometer (Agilent Technologies, Santa Clara, CA, USA), and the oxidation of NADH was monitored using a standard coupled system in which the decay of this cofactor is measured at 340 nm [25]. The results, representing the average of three independent experiments, were normalized as a percentage of enzyme activity, with 100% corresponding to the enzyme activity without drugs.

### 2.5. Quantification of Free Thiols in Recombinant TPI Enzymes Treated with DS or CU

Since it has been established that one of the mechanisms of action of DS and, to some extent, CU is the derivatization of cysteine (Cys) residues, the number of derivatized Cys in the recombinant enzymes was determined using Ellman’s reagent (5,5′-dithiobis-acid (2-nitrobenzoic), DTNB) as previously detailed [26]. TPIs were incubated at 0.2 mg/mL, either in the absence (control with 1.5% ethanol) or in the presence of DS or CU at 250 and 1500 μM, respectively, during 2 h at 37 °C. Following incubation, the samples were extensively washed using ultrafiltration with Centricon filters to remove excess compounds, and the protein concentration was estimated by measuring its absorbance at 280 nm. Subsequently, the content of free thiol (Cys) in the samples was quantified spectrophotometrically as follows: the baseline absorbance of 1 mM DTNB and 5% SDS dissolved in TEA was measured at 412 nm (ε 412 nm  =  14.1 mM^−1^ cm^−1^), and the increase in absorbance was monitored upon addition of 200 µg of protein. The number of derivatized Cys residues was indirectly estimated by subtracting the number of free Cys residues quantified in the derivatized enzyme (+DS or +CU) from the number of free Cys residues quantified in the enzyme in the absence of the drugs, which corresponded to control with 1.5% ethanol. Results are presented as the average of three independent experiments.

### 2.6. Determination of Fluorescence Emission Spectra of TPI Enzymes

The extrinsic fluorescence emission spectra of the proteins were recorded using an LS55 spectrofluorometer (Perkin Elmer, Waltham, MA, USA). The n-dTPI and dTPI enzymes were incubated at 0.2 mg/mL for 2 h at 37 °C, either without control with 1.5% ethanol or with DS and CU at 250 and 1500 μM, respectively. Following incubation, the proteins underwent thorough washing via ultrafiltration with Centricon tubes to remove any residual compounds, and their concentration was determined by measuring its absorbance at 280 nm.

To assess extrinsic fluorescence, 8-anilinonaphthalene-1-sulfonic acid (ANS) dissolved in methanol was prepared. For the experiments, 1 μL of 60 mM ANS was added to 600 μL of the samples, leaving a final concentration of 100 μM. The samples were excited at 395 nm, and fluorescence emission spectra were determined between 400 and 600 nm. For each sample reading, background fluorescence was subtracted, which consisted of TEA with the compounds, TEA with ANS, or TEA with ANS plus DS or CU. The results represent the average of three independent experiments and were normalized as the percentage of fluorescence intensity versus wavelength.

### 2.7. Obtaining Normal T Lymphocytes from Healthy Adults

To conduct comparative assays with Jurkat cells, normal T lymphocytes from healthy donors were used. Peripheral blood samples from three healthy individuals of legal age between 18 and 22 years old were obtained at the National Institute of Pediatrics in Mexico City during the first half of 2023.

The study was approved by the Institutional Research, Biosafety, and Ethics Committees (protocol number: 2022/067), according to the Declaration of Helsinki, and informed consent was obtained from all volunteer donors. Each volunteer donated 50 mL of blood, from which was performed density gradient separation using Lymphoprep™ Density Gradient Medium (STEMCELL Technologies, Germany GmbH, Cologne, Germany). The samples were centrifuged at 2000 rpm for 20 min to isolate the mononuclear cells, including normal T lymphocytes; each sample was resuspended in 1 mL of PBS.

Magnetic nanoparticles conjugated with anti-CD3 antibody (a specific T-lymphocyte marker) from MACS (Miltenyi Biotec, San Diego, CA, USA) were added to the suspension of mononuclear cells. T lymphocytes were then isolated using magnetic separation with the AUTOMACS equipment (Miltenyi Biotec, San Diego, CA, USA). This device processes the entire cell suspension, separating the cells into a positive fraction (T lymphocytes) and a negative fraction (remaining cells).

### 2.8. Cell Cultures and Regimen Treatments with DS, CU, and DCA

The human T lymphoblast cell line Jurkat E6-1 (TIB-152) was obtained from the American Type Culture Collection (ATCC, Rockville, MD, USA) to perform cell culture assays. Jurkat E6-1 cells were cultured in RPMI-1640 medium (Sigma-Aldrich, St. Louis, MO, USA) supplemented with 10% (*v*/*v*) fetal bovine serum, 2 mM L-glutamine, 1 mM sodium pyruvate, 100 U/mL penicillin, and 100 mg/mL streptomycin. Normal T lymphocytes were cultured under similar conditions; RPMI was a suitable culture medium for T-cell expansion following anti-CD3 stimulation.

For all experiments, Jurkat cells from passages 2 to 5 and freshly isolated normal T lymphocytes were utilized. The cells were cultured at 37 °C in a humidified atmosphere with 5% CO_2_. Following culture, the cells were centrifuged at 2500 rpm for 5 min and washed three times with phosphate-buffered saline (PBS).

Jurkat cells and normal T lymphocytes were exposed to increasing concentrations of DS, CU, and DCA to evaluate cell viability and cellular TPI activity. While in all control conditions (in the absence of DS or CU compounds), cells were incubated with the highest percentage of ethanol used (1.5%).

Cells were maintained at a density of 1 × 10^5^ cells per well in a final volume of 1 mL in six-well plates and exposed to the compounds for 24 h under specified culture conditions. The concentrations used were as follows: for DS, 0, 50, 100, 250, 500, and 1000 µM; for CU, 0, 100, 250, 500, 1000, and 1500 µM; and for DCA, 0, 1, 2, 4, 5, 10, and 12 mM. Additionally, preincubation with DCA at 12 mM was performed for 24 h, followed by the addition of DS at 0, 10, 25, 50, 100, and 250 µM or CU at 0, 50, 100, 250, and 500 µM. In all conditions, after the incubation period, the cells were centrifuged, resuspended in PBS, and washed three times.

Then, cell density was determined using a hemocytometer, and cell viability was assessed using the 3-(4,5-dimethylthiazol-2-yl)-2,5-diphenyltetrazolium bromide (MTT) assay. Jurkat cells or normal T lymphocytes (1 × 10^3^) were incubated in 100 μL of PBS/well in a 96-well plate. After the addition of MTT, the cells were incubated for 4 h in the dark. The resulting formazan crystals were dissolved in DMSO, and the absorbance was monitored at 570 nm using an Epoch microplate spectrophotometer (BioTek, Winooski, VT, USA). The results represent the average of three independent experiments and are presented as a percentage of viability relative to the pharmacological treatment, with the viability of untreated cells set at 100%.

To determine cellular TPI activity, a coupled assay system was used as described in Section 2.4. After incubation with the compounds, Jurkat cells or normal T lymphocytes were washed and resuspended in PBS. The cells were then lysed by five freeze/thaw cycles (10 s in liquid nitrogen/1–2 min at 37 °C). Protein concentration was determined using Bradford assays. For enzymatic activity measurement, 40 µg of protein extract was added to a final volume of 500 µL of the reaction mixture, and NADH consumption was monitored spectrophotometrically at 340 nm. The results represent the average of three independent experiments and are expressed as a percentage of enzyme activity, establishing as 100% the activity of cellular TPI of cells without compounds.

### 2.9. Identification of TPI Acidic Isoforms (dTPI) in Cellular Protein Extracts through Native Gel Electrophoresis and Western Blot Analysis

TPI acidic isoforms in cell extracts were identified by native gel electrophoresis (nPAGE) followed by Western blot analysis using an anti-TPI monoclonal antibody (H11) (Appendix A). Jurkat cells or normal T lymphocytes (5 × 10^6^) were treated under the following conditions. Normal T lymphocytes were treated with 0, 250 μM DS, and 1500 μM CU, while Jurkat cells were treated with 0, 100, and 250 μM DS, as well as 0, 500, 1000, and 1500 μM CU, all for 24 h under the previously described conditions. Additionally, a combined treatment was performed where normal T lymphocytes were pre-treated with 12 mM DCA for 24 h, followed by a subsequent 24 h exposure to either 250 μM DS and 1500 μM CU. Similarly, Jurkat cells were pretreated with 12 mM DCA, followed by incubation with either 100 or 250 μM DS, and 1000 or 1500 μM CU.

At the end of the assays, cells were lysed in cold RIPA lysis buffer supplemented with protease inhibitors, and protein quantification was conducted using the Bradford method. nPAGE gels were prepared with 7% polyacrylamide and Tris-glycine buffer at pH 8.5. In each lane, 1 µg of recombinant proteins was loaded, including n-dTPI, dTPI, and ddTPI. Additionally, 100 µg of protein extracts from both Jurkat cells and healthy T lymphocytes were loaded in subsequent lanes.

In a refrigerated environment, samples underwent constant electrophoresis at 7 mA for 3 h. Subsequently, transfer to a polyvinylidene difluoride membrane (PVDF) was carried out at 0.8 mA/cm^2^ for 1 h in 25 mM Tris with 192 mM glycine and 20% methanol. The membrane was then blocked for 1 h with Tris-buffered saline and 0.1% Tween-20 (TBS-T), supplemented with 5% bovine serum albumin (BSA), followed by a single wash with TBS-T. The membrane was then incubated overnight at 4 °C with the anti-TPI antibody (H11) diluted at 1:1000 in TBS-T containing 1% BSA. Subsequently, it was washed three times with the same buffer.

For detection, a horseradish peroxidase (HRP)-conjugated anti-mouse IgG secondary antibody, diluted 1:3000 (Appendix A), was used, and immunoblot bands were revealed by chemiluminescence using Clarity Western ECL substrate (Bio-Rad) according to the supplier’s instructions. Blot image acquisition was performed using a ChemiDoc XRS+ system (Bio-Rad Laboratories, Inc., Hercules, CA, USA). All assays were carried out in triplicate.

### 2.10. Quantification of Methylglyoxal and Advanced Glycation End Products in Jurkat Cells under Treatment Regimen

Quantification of both methylglyoxal (MGO) and advanced glycation end products (AGEs) was conducted using 1 × 10^6^ Jurkat cells treated with 0 and 12 mM DCA, as well as 100 or 250 µM DS and 250 or 500 µM CU. Additionally, the synergistic effect of these compounds was evaluated by incubating cells with 12 mM DCA for 24 h, followed by treatment with 100 or 250 µM DS and 250 or 500 µM CU for an additional 24 h under the same conditions.

Following incubation, cell cultures underwent centrifugation at 2500 rpm for 10 min at 4 °C, and the resulting cell pellet was resuspended in PBS, repeating this step three times. After, cells were resuspended at a density of 5 × 10^6^/200 μL of PBS and lysed through five freeze-thaw cycles (10 s in liquid nitrogen/1–2 min at 37 °C). Aliquots were then withdrawn, and 10% perchloric acid was added to each sample, which was then chilled on ice for 10 min before centrifugation at 12,000 rpm at 4 °C for 10 min. The supernatant was taken and stored at −70 °C until use. Intracellular free MGO was determined spectrophotometrically using 2,4-dinitrophenylhydrazine (DNPH), following the method described by Gilbert and Brandt [27] with modifications [27]. Prior to MGO estimation, a standard curve was established using MGO stock solutions (0.1 mM) in distilled water and 20 mM DNPH in HCl-ethanol (12:88). Various concentrations of MGO (ranging from 0 to 10 μM) were incubated with 0.2 mM DNPH at 42 °C for 45 min. After incubation, samples were allowed to cool for 5 min at room temperature, and the absorbance of MGO-bis-2,4-dinitrophenylhydrazone was measured at 432 nm using a microplate spectrophotometer (Epoch, BioTek, Winooski, VT, USA). Following this, cell supernatants treated with DCA, CU, or their combination were utilized to quantify MGO levels with DNPH-HCl-ethanol. Intracellular free MGO concentrations were estimated from the standard curve, utilizing the extinction coefficient ε = 33,600 M^−1^ cm^−1^ for MGO-bis-2,4-dinitrophenyl-hydrazone. All assays were performed in triplicate, and results are presented as MGO (μM)/1 × 10^6^ cells.

Conversely, the determination of AGEs was conducted using an AGE ELISA kit (MyBioSource, San Diego, CA, USA) as per the manufacturer’s protocol. Aliquots from cell lysates were utilized to determine protein concentrations, which were adjusted to 1 mg/mL, then diluted 1:100 and loaded onto ELISA plates for AGEs concentration determination. Avidin-peroxidase conjugates were added to the wells, and 3,3′,5,5′-tetramethylbenzidine (TMB) was employed as the substrate for staining following thorough washing with PBS. A standard curve was generated using AGEs standards provided in the kit, with concentrations ranging from 0 to 200 ng/mL. Absorbance at 450 nm was measured within the initial 10 min using an Epoch microplate spectrophotometer (Epoch, BioTek, Winooski, VT, USA). Results represent the mean of three independent experiments and are expressed as AGEs (μg/mL).

### 2.11. Western Blot Analysis of Pro- and Anti-Apoptotic Elements

Jurkat cells (5 × 10^6^) were treated with 0 and 12 mM DCA, as well as 500 µM CU. Additionally, the synergistic effect of these compounds was evaluated by incubating cells with 12 mM DCA for 24 h, followed by treatment with 500 µM CU for an additional 24 h under the same conditions. Subsequently, cells were treated as mentioned [28]. Briefly, cells were resuspended and lysed in ice-cold RIPA buffer supplemented with protease inhibitors, with the lysates stored at −70 °C until further use. Protein samples of 100 µg/well were loaded onto 16% SDS-PAGE and initially run at constant voltage (100 V), followed by a subsequent increase to 150 V during 2 h at 4 °C. Then, gels were electroblotted to PVDF membranes, as mentioned in the nPAGE section. After blocked and washed, the membranes were incubated overnight at 4 °C with primary antibodies targeting ERK 1/2 (C-9), p-ERK1/2 (12D4), Caspase-7 (10-1-62), Bcl-2 (C-2), Bax (B-9), and β-Actin (C-2) (Santa Cruz Biotechnology, Santa Cruz, CA, USA) at a dilution of 1:1000 in 0.1% TBS-Tween-20 and 1% BSA (Appendix A). Immediately after, membranes were washed three times with the same buffer. Protein identification was achieved using an HRP-conjugated secondary antibody (diluted 1:3000) and chemiluminescent substrate (Clarity Western ECL substrate, Bio-Rad, Hercules, CA, USA). Blot images were acquired using a Molecular Imager^®^ Gel Doc™ XR + system (Bio-Rad, CA, USA), and the optical density of protein bands was calculated following background subtraction and normalization to β-Actin using Image Studio 4.0 software (LI-COR Biotechnology, Lincoln, NE, USA).

### 2.12. *Flow Cytometry* Apoptosis Assays

Normal T lymphocytes and Jurkat E6-1 cells were cultured in 6-well plates at a density of 1 × 10^6^ cells per well in 2 mL of medium. Cells were treated with 0 and 12 mM DCA, 100 µM DS, and 250 µM CU. To assess potential synergistic effects, cells were first incubated with 12 mM DCA for 24 h, followed by an additional 24 h treatment with 100 µM DS or 250 µM CU under the same conditions. After treatments, cell density was measured using a Neubauer chamber. A control group of cells cultured in 1.5% ethanol was included to evaluate its effect on cell viability.

Following the treatments, cells were stained at a density of 1 × 10^6^ cells/mL with Annexin V and Propidium Iodide to determine the percentages of cells in early apoptosis, late apoptosis, and necrosis. H_2_O_2_-treated samples, at concentrations of 50 µM for 5 h (to induce apoptosis) and 500 µM for 5 h (to induce necrosis), along with untreated/unstained controls, were used as compensatory controls for generating the compensation matrix and subsequent analysis of experimental samples.

Cell analysis was carried out using a Guava^®^ easyCyte™ Flow Cytometer (Cytek^®^ Biosciences, Fremont, CA, USA), with data acquisition and analysis performed using InCyte™ Software v3.1 (Merck Millipore, Bedford, MA, USA).

## 3. Results

### 3.1. In Silico and In Vitro Evaluation of the Recombinant dTPI Model with Repurposed Drugs DS and CU

#### 3.1.1. Deamidated TPI Is Selectively Inactivated by the Repurposing Drugs Disulfiram and Curcumin

Previous studies have demonstrated the increased permeability of deamidated TPI (dTPI) to hydrophobic compounds such as 5,5′-dithiobis-(2-nitrobenzoic acid) (DTNB) and 8-anilinonaphthalene-1-sulfonic acid (ANS) [9,28], respectively. Thus, this study explored the potential for selective dTPI inhibition using similar compounds. A molecular docking analysis was conducted, utilizing crystallographic structures of both n-dTPI and dTPI to assess the selectivity of the cysteine-modifying drug DS and the naturally occurring hydrophobic CU.

The obtained results indicate that, unlike other binding sites, a higher binding affinity of DS to the dimeric interface of dTPI compared to n-dTPI was observed (Figure 1a,b and Appendix A). This observation aligns with the substantial difference in interfacial volume between the deamidated (287.4 Å^3^) and non-deamidated (105.6 Å^3^) enzymes, suggesting greater accessibility of DS to nearby Cys residues within the dTPI interface. Furthermore, molecular docking analyses also demonstrated enhanced accessibility of CU to the dTPI interface compared to the n-dTPI interface (Figure 1c,d). This increased accessibility can be attributed to the presence of broader hydrophobic regions within the dTPI interface, facilitating deeper penetration of CU into less solvent-exposed cavities.

Both DS and CU docked at the n-dTPI interface and other regions (Appendix A); however, their binding affinities were weaker compared to those observed for the deamidated form (Appendix A). Moreover, examination revealed prominent hydrophobic cavities at both the entrance and interior regions of the dTPI interface, further supporting the preferential binding of CU to this region (Figure 1c,d). Consequently, the porous dimeric structure of the deamidated protein likely facilitates access and binding of both DS and CU at its interface, potentially leading to selective inhibition.

To validate the in silico docking findings, enzymatic activity assays were conducted with recombinant n-dTPI and dTPI enzymes, both in the presence and absence of DS and CU. The results confirmed the computational predictions, revealing a gradual decrease in dTPI enzyme activity with increasing concentrations of both compounds (Figure 2a,b). Contrastingly, n-dTPI activity remained largely unaffected compared to the control (absence of DS or CU, with 100% of enzyme activity) even at concentrations as high as 1000 μM (Figure 2a,b, black squares). dTPI enzyme activity exhibited a clear and consistent decline with increasing concentrations of both DS and CU (Figure 2a,b, red squares). For example, at 250 μM DS, its activity was completely abolished, while CU displayed a more gradual and less pronounced inhibitory effect, reaching approximately 90% inhibition at 1500 μM.

Based on the molecular docking results demonstrating that both DS and CU preferentially bind to the dTPI interface, additional experiments were performed to investigate the sequential inactivation of dTPI by these compounds. In such experiments, DS was incubated with dTPI first, followed by the addition of CU, or vice versa. The results indicated that the inactivation was primarily driven by the first compound introduced, suggesting that there may be competition for the same binding pocket. When the first compound occupies the pocket, it could limit the access of the second compound. These results may imply that while both compounds interact with similar regions of the enzyme, their binding sites could differ, or the mechanisms of inactivation might vary depending on the sequence of administration.

The findings from both molecular docking simulations and enzymatic activity assays strongly support the hypothesis of high selectivity for dTPI in both DS and CU, resulting in a significant impairment of its enzymatic activity. This observed difference in dTPI inhibition between the deamidated and non-deamidated forms suggests the possible therapeutic application of DS and CU against this specific TPI isoform, considering that it may be enriched in cancer cells.

To elucidate the potential mechanisms underlying the observed loss of enzymatic activity in the deamidated enzyme, we investigated the derivatization of Cys residues by DS and, to a lesser extent, by CU, known for their ability to chemically modify such residues in proteins [29]. Recombinant n-dTPI and dTPI enzymes were incubated at 37 °C for 2 h, either without any treatment or with 250 μM DS or 1500 μM CU. After incubation, excess DS or CU was removed by washing with Centricon tubes. The remaining free Cys residues were then quantified under denaturing conditions based on the Ellman method [26].

In Table 1, it is shown that both n-dTPI and dTPI were quantified at approximately 5 free Cys residues per subunit without DS or CU treatment. This observation aligns with the five Cys residues reported in the primary structure of TPI (NCBI Reference Sequence: NP_000356.1), confirming the reactivity of these aminoacyl residues. Following incubation with DS, n-dTPI retained approximately 4 Cys residues per subunit (Table 1). This strongly suggests the successful derivatization of 1 Cys by DS, which notably did not significantly affect enzymatic activity (Figure 2a, black squares). Conversely, exposure of dTPI to DS drastically decreased in the number of detectable free Cys residues, dropping to only 2 per subunit. This marked difference implies the derivatization of 3 Cys residues per subunit by DS, leading to complete enzyme inactivation (Figure 2a, red squares).

On the other hand, when TPIs were exposed to CU, no derivatized Cys residues were registered. This strongly suggests that the mechanism of inactivation observed in dTPI with CU does not involve the derivatization of its Cys residues.

Finally, the results with DS suggest a critical threshold effect. Derivatization of three or more Cys residues in the deamidated form appears necessary for complete loss of its enzymatic activity (Figure 2a). This reinforces the potential of thiol-reactive drugs, such as DS, for the selective inactivation of proteins, particularly those containing multiple solvent-accessible Cys residues [28].

#### 3.1.2. DS and CU Promote Structural Alterations in dTPI

Since DS and CU interactions with proteins can induce functional alterations [30,31], we employed extrinsic fluorescence spectroscopy to investigate the structural changes induced in TPI upon exposure to these compounds. The hydrophobic fluorescent probe ANS was utilized to probe solvent-exposed hydrophobic cavities within the protein structure. Following incubation of n-dTPI and dTPI enzymes with DS and CU at concentrations corresponding to the highest observed enzymatic inactivation (Figure 2), extrinsic fluorescence emission spectra were measured. Excitation was set at 395 nm, and emission was monitored between 400 and 600 nm.

The results revealed a remarkable tenfold increase in the fluorescence intensity of dTPI compared to n-dTPI in the presence of ANS but in the absence of DS and CU (Figure 3a). This observation suggests enhanced accessibility of ANS to hydrophobic cavities within the deamidated enzyme, compared to the more restricted access in the non-deamidated form. This finding aligns with the increased binding affinity of both DS and CU towards dTPI observed in the docking simulations (Figure 1).

Following exposure to DS or CU, a significant decrease in extrinsic fluorescence was observed, particularly in dTPI (Figure 3b,c). Both enzymes exhibited fluorescence values close to 10% of the untreated dTPI value (Figure 3a). Importantly, dTPI displayed a remarkable 90% reduction in extrinsic fluorescence upon incubation with either DS or CU (Figure 3b,c red lines). These results suggest that ANS has diminished access to hydrophobic cavities within the deamidated TPI when these compounds are present.

This observed decrease in fluorescence intensity likely reflects the occupancy of these hydrophobic cavities by DS or CU. Strongly suggesting a high affinity of both compounds for solvent-exposed hydrophobic regions, particularly those located within the dimeric interface. In general, the obtained results from the recombinant enzymes are consistent with the findings from the molecular docking simulations, which demonstrated preferential binding of DS and CU to such hydrophobic zones within the deamidated enzyme (Figure 1).

At the computational and recombinant level, a clear distinction between n-dTPI and dTPI is demonstrated. Deamidated TPI exhibits increased susceptibility to functional and structural alterations upon exposure to DS and CU. This vulnerability underlines the potential of dTPI as a selective therapeutic target in cancer cells. Furthermore, the observed effects of DS and CU on dTPI activity and structure support the exploration of these compounds as potential anticancer agents, particularly those targeting tumors enriched in the deamidated TPI isoform.

### 3.2. Evaluation of the Cellular Efficacy of DS and CU and Determination of the Presence of dTPI

#### 3.2.1. Jurkat Cells Are Selectively Sensitive to DS and CU

Taking advantage of the selective inhibition by DS and CU observed in recombinant dTPI (Figure 2), we investigated their potential efficacy in a cellular context. Jurkat cells were treated with increasing concentrations of these compounds to assess their impact on both cell proliferation and endogenous TPI activity. In parallel, similar experiments were conducted on normal T lymphocytes to compare the effects of these compounds under the same conditions.

Jurkat and normal T lymphocytes (1 × 10^5^ cells/well) were incubated separately with increasing concentrations of DS and CU for 24 h at 37 °C in a 5% CO_2_ atmosphere. Following incubation, cell viability and cellular TPI activity were quantified.

In normal T lymphocytes, cell viability and TPI activity remained near 100% at all concentrations tested with DS and CU, with no significant differences compared to the untreated control (Figure 4a,c).

In contrast, Jurkat cells exhibited a drastic decrease in cell viability and TPI activity upon exposure to both compounds. Specifically, cell viability and endogenous TPI activity showed a progressive decline, ultimately leading to complete inhibition at higher concentrations (Figure 4b,d). Notably, a significant reduction in TPI activity was observed with as low as 50 μM DS, while a slightly higher concentration (100 μM) was required to produce a significant decrease in cell viability (Figure 4b). As the concentration increased, both parameters gradually decreased, with a more pronounced effect on TPI activity. Thus, DS demonstrated an IC_50_ of 454 μM for cell viability (Appendix A). In comparison, CU exhibited a less potent inhibitory effect, with both cell viability and TPI activity gradually decreasing, reaching significance at concentrations starting from 500 μM (Figure 4d). A concentration of 1500 μM CU was required to reduce both cell viability and TPI activity by more than 80%, with an IC_50_ of 829 μM for cell viability (Appendix A), further highlighting the greater effectiveness of DS.

Our data demonstrates significant selectivity of DS and CU for Jurkat cells. These results strongly suggest the therapeutic potential of DS and CU as selective anti-cancer agents by targeting TPI. Furthermore, the differential susceptibility of TPI between normal and cancer cells highlights potential intrinsic structural or functional differences that warrant further investigation.

#### 3.2.2. Treatment with DCA Contributes to Reducing the Dose of DS or CU Required to Inhibit Jurkat Cells

Given the pronounced effects of DS on Jurkat cell viability and TPI activity compared to the milder effects observed with CU, we explored the potential of DCA as a synergistic agent. Previous studies have shown that DCA can enhance the cytotoxicity of diverse anticancer drugs, making it a promising candidate to boost the efficacy of DS and CU [32,33,34]. Therefore, we included DCA as a pretreatment to increase the efficacy of DS and CU against Jurkat cells. Thus, to evaluate the effects of DCA on cellular viability and TPI activity, both normal T lymphocytes and Jurkat cells were exposed to increasing concentrations of DS and CU.

As shown in Appendix A, treatment with DCA at concentrations up to 10 mM did not significantly affect cell viability or TPI activity in normal or cancer cells compared to the control without compounds. Although later a slight decrease of approximately 10% in both parameters was observed only at the highest concentration tested (12 mM). These results demonstrate that, although Jurkat cells are slightly sensitive (approximately 10% of inhibition), DCA treatment at these concentrations is well-tolerated by both cell types.

To evaluate the potential for DCA-mediated sensitization, Jurkat cells were pre-treated with 12 mM DCA for 24 h, followed by additional 24 h incubation with increasing concentrations of DS (0–250 μM) or CU (0–500 μM) (Figure 5). In normal T lymphocytes pre-treated with DCA, subsequent treatment with DS or CU had minimal effects on cell viability and cellular TPI activity (Figure 5a,c, respectively). In contrast, DCA-pretreated Jurkat cells showed a marked reduction in cell viability and endogenous TPI activity when exposed to DS or CU (Figure 5b,d, respectively). This combined treatment reduced the concentrations of DS and CU required to achieve effective anti-cancer effects. Specifically, the IC_50_ of the DCA+DS combination was 123 μM, and the IC_50_ of the DCA+CU combination was 300 μM (Appendix A). Comparison of the IC_50_ values for DS and CU treatments alone with those in combination with DCA clearly demonstrates that DCA significantly enhances the cytotoxic efficacy of both compounds. The combination of DCA with DS resulted in a 3.7-fold reduction in the concentration needed to achieve the same cytotoxic effect, while the combination with CU led to a 2.8-fold decrease in the effective concentration. These findings highlight DCA’s potent sensitizing effect, making both DS and CU considerably more effective at reduced concentrations when used in combination with DCA.

#### 3.2.3. nPAGE Analysis Reveals Significant Differences in the Composition of Acidic Isoforms of TPI between Jurkat Cells and Normal T-Cell Lymphocytes

Previous studies have confirmed the presence of acidic isoforms of TPI in mitogen-stimulated lymphoblasts, attributing the multiple variants revealed on native gels (nPAGE) to specific deamidation of asparagine residues of TPI at aminoacyl positions 16 and 72 [35]. Based on this, we hypothesized that these isoforms might also be present in Jurkat cells. To investigate this, we analyzed the TPI isoform composition in both normal T lymphocytes (untreated or treated with DS or CU) and Jurkat cells (untreated or treated with DCA, DS, or CU). Additionally, Jurkat cells were pre-treated with DCA followed by DS or CU incubation, as described in the Methods section. This allowed us to evaluate the impact of each treatment and combination on the TPI isoform profile in these cell types. Cells were incubated with the respective compounds, lysed, and protein extracts were quantified for subsequent separation using native polyacrylamide gel electrophoresis (nPAGE). Recombinant TPI isoforms served as migration reference standards. Following electrophoresis, recombinant enzymes and total cell lysates were transferred to membranes and analyzed by Western blot using an anti-TPI antibody.

As shown in Figure 6a,b, lanes 1, 2, and 3 correspond to the migration pattern of the recombinant TPI isoforms on the nPAGE: non-deamidated (n-dTPI), single-deamidated isoform (dTPI), and double-deamidated isoform (ddTPI), respectively. Due to the mass-to-charge ratio dependence of protein migration in nPAGE, dTPI, and ddTPI exhibit a distinct anodal shift compared to n-dTPI due to their higher net negative charge.

Analysis of the blots reveals contrasting differences in TPI isoform composition between normal T-lymphocyte and Jurkat cells, with these profiles further influenced by DCA, DS, or CU treatment. A single band corresponding primarily to n-dTPI was observed in normal T lymphocytes under all conditions (control, as well as DS or CU treatment; Figure 6a, lanes 4–6, respectively). This strongly suggests the absence of acidic isoforms in these normal cells. Importantly, Jurkat cells displayed a more complex isoform profile, harboring n-dTPI and acidic isoforms (dTPI and ddTPI) (Figure 6a,b, lanes 7–9 and 4–9, respectively). The control (lanes 7 and 4, in 6a,b, respectively) show two bands, one migrating at level of n-dTPI and the other at level of dTPI. Notably, treatment with DS and CU (lanes 8 and 9 in a) resulted in an increase in dTPI and the appearance of ddTPI. Furthermore, treatment of Jurkat cells with DCA or increasing concentrations of DS or CU resulted in increased dTPI and appearance of ddTPI (Figure 6b, lane 5 and lanes 6–7 and 8–10, respectively).

These observations are consistent with previous reports demonstrating the presence of acidic isoforms corresponding to dTPI in cancer cells such as the aggressive breast cancer cell line MDA-MB-231 while being absent in their normal cell counterparts [28]. The detection of these acidic isoforms provides valuable insights into the unique characteristics of cancer cells and their response to specific treatments, including those involving DCA, DS, or CU.

Additional nPAGE and Western blot analyses were performed on protein extracts from cells pretreated with DCA followed by DS or CU treatment to further validate these observations. Consistent with the previous results, no acidic TPI isoforms were detected in normal T lymphocytes, despite pre-incubation with DCA combined with DS or CU, reaffirming their absence even under synergistic pharmacological treatment (Figure 7a,b, lane 9). In contrast, untreated Jurkat cells displayed two distinct bands corresponding to n-dTPI and dTPI (Figure 7a,b, lane 4). Importantly, pretreatment with DCA specifically increased the deamidated TPI band (Figure 7a,b, lane 5), whereas combined treatment with DCA with either DS or CU further enriched the deamidated TPI isoforms (Figure 7a,b, lanes 6 and 7).

Finally, as individual treatments or pretreatments induced the appearance of distinct TPI isoforms in Jurkat cells, their relative abundance (respect to the isoform n-dTPI of each lane) was analyzed using densitometry and is summarized in Appendix A. In untreated cells, n-dTPI is the dominant form, with only a small percentage of dTPI (~10%) and no detectable ddTPI. When treated with 12 mM DCA, the level of dTPI increases significantly to ~90%, but ddTPI remains undetected. With 250 μM DS, both dTPI and ddTPI levels rise, reaching ~90% and ~10%, respectively. The treatment with 1500 μM CU causes a further increase in dTPI to ~150%, with ddTPI at ~80%. The most pronounced effects are observed when combining DCA with either DS or CU. The combination of DCA with DS results in a notable rise in both dTPI and ddTPI by more than 700%. Similarly, the combination of DCA and CU boosts dTPI to ~700% and ddTPI to ~600%.

These findings demonstrate the ability of DCA, DS, and CU treatments to selectively enrich deamidated TPI isoforms in Jurkat cells. This enrichment process potentially renders cancer cells more susceptible by increasing the pool of drug-sensitive TPI variants, especially in combined treatments (DCA-DS and DCA-CU). Therefore, our observations highlight the potential of targeting these specific TPI isoforms for the development of novel therapeutic strategies against T-ALL.

#### 3.2.4. Inactivation of TPI in Jurkat Cells Promotes the Generation of Methylglyoxal and Advanced Glycation End-Products

Inactivation of TPI is known to trigger a cascade of cellular events. TPI deficiency leads to the accumulation of its substrates, ultimately resulting in its degradation and the generation of the dicarbonyl aldehyde methylglyoxal (MGO). MGO is primarily formed as a byproduct of glycolysis [36]. MGO reacts with aminoacyl residues of proteins and nucleic acids, leading to the formation of Advanced Glycation End-Products (AGEs) via glycation and autooxidation [37].

Based on the observation that pretreatment with DCA plus DS or CU inhibits TPI activity in Jurkat cells, we investigated the levels of MGO and AGEs following single or combined treatments. Jurkat cells were incubated with DCA, DS, or CU, or their combination, as detailed in the Methods section. After incubation, cells were washed and lysed, and protein removal was performed with perchloric acid. MGO concentration was then determined using a modified Gilbert and Brandt method [27] described by [28].

Our results revealed a basal MGO concentration of 90 nM/million cells in untreated Jurkat cells (Figure 8). This aligns with previous studies suggesting that cancer cell metabolism can induce a hormetic effect of MGO, contributing to the cancerous phenotype [38]. Treatment with 12 mM DCA resulted in a near 5.3-fold increase in MGO concentration compared to controls. Notably, incubations with 100 and 250 μM DS resulted in a more substantial rise with 3-fold and 5-fold increases, respectively (Figure 8a). The combination of 12 mM DCA followed by 100 or 250 μM DS led to drastic increases in MGO leves (Figure 8a). Regarding the determination of AGEs, the results were similar to those of MGO, that is, with respect to the untreated control, they were proportionally increased with the treatments alone or in combination with DCA (Figure 8b).

Treatment with CU produced results comparable in magnitude to those observed with DS. That is, 250 μM or 500 μM CU led to substantial increases in MGO concentration, with 10-fold and 18.7-fold increases, respectively (Figure 8c). When 12 mM DCA was combined with 250 μM or 500 μM CU, the increases in MGO concentration were even more pronounced, reaching 31-fold and 57-fold, compared to the untreated control (Figure 8c). These findings were further supported by the proportional increase in AGEs levels relative to the untreated controls (Figure 8d). Thus, our data suggest that TPI inactivation in Jurkat cells triggers a cascade of events that promotes MGO generation and subsequent AGEs formation, which may ultimately tigger apoptotic mechanisms [39].

#### 3.2.5. Jurkat Cells Exposed to DS, CU, or Combined with DCA Undergo Apoptosis

Based on the observed MGO and AGEs accumulation, we investigated the potential for these events to trigger apoptosis in Jurkat cells. MGO is a well-established inducer of apoptosis, promoting activation of pro-apoptotic signaling pathways (e.g., caspases and Bax) while inhibiting anti-apoptotic pathways (e.g., ERK1/2 phosphorylation) [40]. Consistent with this established role of MGO, we sought to elucidate the influence of DCA and CU treatments on key apoptotic regulators, potentially opening new avenues for cancer research.

Several anti-cancer agents are known to exert their effects through MGO and AGEs overproduction, ultimately triggering apoptosis [41]. To further explore the potential for DCA and CU to induce apoptosis in Jurkat cells, the expression levels of proteins associated with intrinsic apoptotic pathways were analyzed, specifically the pro-apoptotic Bax and the anti-apoptotic Bcl-2.

Our results revealed that treatment with DCA, CU, or their combination caused a progressive reduction in the expression of both total and phosphorylated ERK1/2 (Figure 9a). This was accompanied by a decrease in Bcl-2 expression (Figure 9b), while pro-apoptotic markers Bax (Figure 9b) and cleaved procaspase-7 (Figure 9c) showed a significant increase following treatment.

To reinforce the previous results and to demonstrate possible apoptotic events in normal cells, flow cytometry analyses were performed in Jurkat cells and normal T lymphocytes.

Flow cytometry analysis was performed using Annexin V and propidium iodide staining, established markers for distinguishing between early and late apoptosis and necrosis, as confirmed by the apoptosis and necrosis controls in Appendix A.

The flow cytometry results reveal a clear distinction in the response to drug treatments between normal T lymphocytes and Jurkat cells. In normal T lymphocytes (Figure 10 and Appendix A), apoptosis is minimal under all conditions, with cell viability remaining high (84–97%) in the control (without treatment). Even when treated with DCA, DS, or CU alone or in combination, early and late apoptosis percentages remain low, in general below 5%, indicating that these treatments do not induce cell death in normal cells. Necrosis levels also stay relatively constant and low.

In contrast, Jurkat cells (Figure 10 and Appendix A) show a notable increase in apoptosis and necrosis following treatment. While the control group shows minimal apoptosis (1.08% early, 0.26% late), treatment with DCA, DS, or CU leads to a marked rise in both early and late apoptosis, especially when the drugs are combined. The combination treatments (DCA-DS or DCA-CU) induce the highest levels of apoptosis, with early and late apoptosis reaching 8.13% and 19.54%, respectively. Correspondingly, viability in Jurkat cells drops with combined treatments, falling to around 57% compared to 98% in the control group.

Taken together, our data demonstrate the potential of DS, CU, and DCA as effective agents that promote apoptosis in cancer cells, as demonstrated by the marked increase in early and late apoptosis and the substantial decrease in cell viability. The synergistic effect strongly suggests that the combined treatment is more effective in inducing apoptosis than either agent alone, highlighting the potential therapeutic value of this combination to induce apoptosis in such cancer cells.

## 4. Discussion

A cornerstone of contemporary oncology research revolves around identifying highly selective protein targets within cancer cells. This strategy capitalizes on the distinct protein profiles of cancer cells compared to their healthy counterparts [42]. While inherently complex and challenging, advancements in research have unveiled significant metabolic and protein overexpression disparities between these cell types [43].

A compelling example of this approach is the human epidermal growth factor receptor 2 (HER2), which is overexpressed in specific breast cancers but not normal cells [44]. Such condition renders HER2 an ideal target for targeted therapies. Consequently, drugs such as trastuzumab have been developed to block the receiver function of HER2 in cancer cells, demonstrating significant efficacy and safety [45]. Targeted therapies, exemplified by trastuzumab, epitomize the immense potential of exploiting unique protein expression profiles to design more effective cancer treatments.

Another substantial difference that may be present in cancer cells is the Warburg effect, which is characterized by the upregulated reliance on aerobic glycolysis over oxidative phosphorylation [46]. While less efficient in ATP generation, this metabolic shift allows cancer cells to prioritize the rapid production of biosynthetic intermediates for nucleic acid, protein, and lipid synthesis, fueling their uncontrolled proliferation [47]. Therefore, this altered glycolytic phenotype creates a vulnerability that can be exploited for therapeutic purposes. Consequently, enzymes within the glycolytic pathway, such as TPI, critical to this altered metabolism, have emerged as attractive targets for anticancer drug development. In this sense, recent studies have identified deamidated TPI variants in different cancers [28,48]. These deamidated isoforms exhibit altered biochemical properties, including increased susceptibility to specific compounds. Our findings provide compelling evidence for the selective inactivation of deamidated TPI by the repurposed drugs DS and CU. This is evidenced by the selective targeting of recombinant deamidated TPI and its endogenous counterpart in T-ALL cells.

Based on previous research demonstrating that deamidation of the Asn at position 16 of human TPI alters its functional and structural properties [28], we investigated the potential for these modifications that contribute to drug binding. Specifically, deamidation has been shown to promote the susceptibility of TPI to small hydrophobic molecules. Our in silico analyses support this hypothesis, demonstrating that both drugs (DS and CU) exhibit a greater affinity and selectivity for the dimeric interface of deamidated TPI compared to their non-deamidated counterpart. Particularly through this interfacial cavity, it is highly probable that DS accesses buried Cys residues.

These findings are corroborated by existing literature. For example, DS exhibits binding affinity for the CRBD structural domain of the chemokine signal regulator FROUNT. This domain is characterized by its hydrophobic features. Upon substitution of the Cys 603 residue with the polar amino acid residue Serine, DS no longer demonstrates significant binding to this cavity [49].

In another study, it was demonstrated that DS can inhibit SARS-CoV-2 infection by disrupting the interaction between the ACE2 receptor and spike proteins. To elucidate how DS interferes with and prevents the binding between ACE2 and both the wild-type and variants of the spike protein, the authors analyzed one hundred docking poses of DS. They identified that the binding between DS and pockets of the wild type and other variants is predominantly hydrophobic, with an average computational binding affinity of −5.2 kcal/mol. Their molecular docking analyses concluded that DS is effective in preventing the binding of both the wild-type and variant spike proteins to ACE2 [50]. Therefore, DS exhibits a binding preference for hydrophobic cavities within protein structures, highlighting the importance of the hydrophobic nature of these pockets for its binding affinity. However, alterations of the hydrophobic environment can significantly reduce or eliminate DS binding, reinforcing the high selectivity of this drug to certain protein targets. Moreover, DS can effectively disrupt protein-protein interactions by targeting hydrophobic pockets, as demonstrated by its ability to prevent the interaction between the ACE2 receptor and SARS-CoV-2 spike proteins. These reports agree with our results, in which it was shown that DS has a greater affinity for the hydrophobic interface of dTPI, inducing its inactivation.

Based on the established importance of the TPI dimeric interface for structural integrity, our study investigated its exploitability for selective targeting of deamidated TPI. Prior research has highlighted the vulnerability of this region in TPIs from *Trypanosoma brucei* and *T. cruzi*. Studies by Pérez-Montfort et al. [51] demonstrated that derivatization of a crucial interfacial Cys residue with sulfhydryl reagents in *T. brucei* and *T. cruzi* TPIs led to progressive structural destabilization and diminished enzyme activity. Similarly, Téllez-Valencia et al. [52] reported the successful suppression of *T. cruzi* TPI activity through its dimeric interface using aromatic heterocyclic hydrophobic compounds derived from benzothiazole. These findings emphasize the potential of targeting the TPI interface with thiol-reactive compounds or hydrophobic molecules to achieve selective enzyme inactivation. Our data strongly suggest that the deamidated human TPI shares this susceptibility, which can be exploited to identify novel molecules that can disrupt its function by targeting the altered interface.

Taken together, the abilities of the compounds and the vulnerabilities of dTPI were further supported with our activity assays. DS completely inactivated the deamidated enzyme at lower concentrations than CU, although CU also exhibited near-complete inhibition. Notably, the enzymatic activity of non-deamidated TPI remained unaffected by these compounds even at the highest tested concentrations. The quantification of derivatized Cys residues further supported this selective inhibition. DS modified a greater number of Cys in deamidated TPI, which correlated with the complete loss of its enzymatic activity, whereas no Cys derivatization was identified in CU-exposed TPI.

Cysteinyl targeting for TPI inactivation is a well-established strategy across diverse organisms. Studies by Hernández-Alcántara, G. et al. [53] exemplified this approach by demonstrating complete abrogation of *Giardia lamblia* TPI activity through the derivatization of its Cys 222 residue. Similarly, Enríquez-Flores et al. [54] reported the inactivation of human deamidated TPI by omeprazole, which modifies specific Cys residues. These findings solidify the effectiveness of Cys derivatization for TPI inhibition, particularly with the potential for selective targeting of specific isoforms. This strategy holds significant promise for the development of targeted therapies, especially in cancers where deamidated TPI or other TPI isoforms play a critical role in fueling the altered metabolic demands of cancer cells. The vast repertoire of protein isoforms generated within the tumor microenvironment further underscores the potential for this approach.

To gain deeper mechanistic insights into the action of DS and CU, we employed extrinsic fluorescence spectroscopy with ANS as a hydrophobic probe. This technique monitors changes in the protein’s tertiary structure by exploiting the sensitivity of ANS fluorescence to its surrounding environment. Our results suggest that the significant decrease in ANS fluorescence intensity following treatment with DS and CU is due to these compounds occupying the hydrophobic cavities in deamidated TPI. This binding likely prevents the ANS probe from accessing these areas. As a result, the functional inactivation observed in enzymatic assays is probably due to the structural disruption caused by DS and CU at the deamidated TPI interface.

The high affinity of DS and CU towards the hydrophobic regions, particularly within the deamidated TPI dimeric interface, underscores their potential for selective targeting of this specific isoform in cancer cells. In silico analyses using the CavityPlus 2022 [55] further corroborate these observations by revealing distinctive cavities between the deamidated and non-deamidated TPIs. This information can be leveraged to rationalize even more selective inhibitors. CavityPlus, a server identifying and ranking potential ligand binding sites within protein cavities, classified the deamidated TPI interface as highly druggable (Appendix A). These findings pinpoint critical “hot spots” in the deamidated form, further supporting its viability as a target for therapeutic development.

In the cellular context, this study emphasizes the potential therapeutic value of DS, CU, and DCA in leukemia treatment, particularly in Jurkat cells. While DS exhibited a pronounced effect on cell viability and TPI activity, our results demonstrate a synergistic effect between DCA-DS and DCA-CU. Specifically, DCA significantly reduced the effective dose of DS and CU required for achieving anticancer effects. This synergistic action likely stems from a combination of mechanisms, including the generation of TPI isoforms, increased MGO production, and subsequent induction of apoptosis in cancer cells.

The ability of DCA to sensitize cancer cells to other therapies has been documented across various malignancies. For instance, Michelakis et al. [56] demonstrated that DCA effectively inhibits the growth of glioblastoma cells by targeting mitochondrial function, ultimately leading to increased apoptosis. Similarly, Sanchez et al. [57] reported that DCA can reverse the Warburg effect, thereby enhancing chemotherapeutic agents’ efficacy. Our findings align with these studies, suggesting that DCA potentiates the effects of CU by promoting analogous metabolic and apoptotic pathways within Jurkat cells. Our observations align with emerging research emphasizing the importance of metabolic reprogramming in cancer therapy for enhanced efficacy and reduced toxicity [58].

Furthermore, this work sheds light on the role of deamidated TPI isoforms in cancer cell metabolism and treatment sensitivity. The exclusive presence of acidic TPI isoforms (dTPI and ddTPI) in Jurkat cells indicates a clear difference that translates into metabolic vulnerability in these cancer cells. DCA treatment further induced and accumulated these isoforms, potentially rendering cancer cells more susceptible to both DS and CU. Similar observations in aggressive breast cancer cell lines (e.g., MDA-MB-231) add weight to the notion of acidic TPI isoforms as promising therapeutic targets [28]. The ability to selectively target these isoforms in cancer cells presents an interesting avenue for developing more precise and effective cancer treatments.

Our findings regarding the increase in MGO levels and subsequent formation of AGEs in DCA-DS and DCA-CU-treated Jurkat cells align with established literature demonstrating a well-defined link between TPI inhibition, MGO accumulation, and apoptosis [41,59]. MGO is a potent inducer of apoptosis, known to modify proteins and nucleic acids, ultimately leading to cellular dysfunction and death [60]. Our data reveal a significant increase in both MGO and AGEs following treatments with drugs, which coincides with the upregulation of pro-apoptotic markers Bax and cleaved procaspase-7 and a concomitant downregulation of the anti-apoptotic protein Bcl-2. This mechanism of action is further supported by studies in epithelial malignancies such as breast and colon cancer, where MGO-mediated apoptosis has been identified as a critical factor in these events [61,62].

Furthermore, similar observations in colorectal cancer cells, where DCA promotes apoptosis specifically in cancer cells [63], strengthen the concept of MGO-induced apoptosis as a common pathway exploited by DCA-based therapies across various cancers. This selectivity for cancer cells further strengthens the therapeutic potential of DCA. A key finding of this study is the induction of apoptosis in Jurkat cells treated with the combination of DCA-DS and DCA-CU. The observed decrease in ERK 1/2 phosphorylation and Bcl-2 protein levels, coupled with the upregulation of pro-apoptotic markers Bax and cleaved procaspase-7, strongly suggests activation of the intrinsic apoptotic pathway by these treatments. Flow cytometry data convincingly corroborate these findings, revealing a significant increase in apoptotic cell populations, particularly following combined DCA with DS or CU treatment. This mechanism aligns with previous observations by Zhou et al. [64], where DCA-mediated apoptosis in ovarian cancer cells proceeded through the mitochondrial apoptotic pathway and involved caspase activation. Furthermore, studies in MCF-7 cells (a human breast cancer cell line) have demonstrated similar results. C9 + DCA treatment in MCF-7 cells was found to induce apoptosis through the intrinsic pathway [65]. These additional examples strengthen the notion that DCA can activate the intrinsic apoptotic pathway across different cancer types, suggesting a translatable therapeutic strategy. Therefore, the capacity of DCA to potentiate the pro-apoptotic effects of DS or CU presents a promising approach to target metabolic vulnerabilities in leukemia cells. This strategy has the potential to improve treatment efficacy while minimizing associated toxicity, offering a valuable avenue for future cancer therapy development.

Drug repurposing, also known as drug repositioning, offers a powerful strategy for uncovering novel therapeutic applications for existing medications beyond their originally intended use. This approach capitalizes on the established pharmacological profiles and safety data of approved drugs, thereby streamlining the development of new treatments and significantly reducing the time and financial resources required to bring new drugs to market. Our findings on the effects of DS and CU as well as the synergistic cytotoxicity observed with the combined treatment of DCA plus DS and DCA plus CU in a model cell of leukemia exemplify the potential of drug repurposing in oncology. This approach holds significant promise for markedly improving the efficacy of cancer treatment regimens.

DCA, initially developed for managing metabolic disorders such as congenital lactic acidosis, has emerged as a promising candidate with anticancer properties. Such a compound acts by targeting mitochondrial metabolism through pyruvate dehydrogenase kinase (PDK) inhibition. This promotes the switch from aerobic glycolysis to oxidative phosphorylation, a metabolic shift that can sensitize cancer cells to apoptosis. Our study demonstrates that DCA effectively enhances the susceptibility of Jurkat cells to the chemotherapeutic agents DS and CU, enabling the use of reduced doses while achieving significant anticancer effects. This successful repurposing of DCA from a treatment for metabolic disorders to an adjunct cancer therapy highlights its versatility in targeting dysregulated metabolic pathways, a hallmark of many cancers, including T-ALL. Beyond leukemia, DCA’s potential as a repurposed therapy extends to various other cancer types; promising results have been demonstrated in Glioblastoma multiforme, a highly aggressive brain tumor [66]. Additionally, breast cancer research explores DCA’s potential to target cancer stem cells, a critical subpopulation for tumor initiation and resistance to therapy. Studies have shown that DCA can reduce the viability and self-renewal capacity of embryonal carcinoma cells [67], highlighting its potential to overcome treatment resistance. These examples showcase the wider applicability of drug repurposing with DCA, extending beyond our specific findings in leukemia.

DS, traditionally used to treat chronic alcoholism, has emerged as a promising candidate for cancer therapy due to its multifaceted mechanisms of action. DS targets key proteins, such as the proteasomal de-ubiquitinating enzyme POH1, crucial for the proteasome’s de-ubiquitination activity. By inhibiting this process, DS disrupts proteasomal function, impairing NF-κB activation—a pathway involved in anti-apoptotic, cancer-promoting, and cell-cycle regulation [68]. Additionally, DS inhibits AKT phosphorylation, disrupting critical survival pathways in tumor cells and further promoting apoptosis [13].

Notably, our pioneering study reveals that combining DS with DCA significantly amplifies its selective anti-cancer effects. This combination enhances inhibition of cell proliferation, impairs TPI enzymatic activity, and elevates the production of MGO and AGEs in Jurkat cells, ultimately leading to increased apoptosis. These findings position DS as a potential adjuvant therapy in leukemia, illustrating the potential of drug repurposing to accelerate cancer treatment development. Leveraging the established safety profiles of drugs such as DS offers a faster, more cost-effective pathway compared to traditional de novo drug discovery, making this approach highly advantageous in the field of oncology.

CU, a well-characterized polyphenolic compound isolated from *Curcuma longa*, boasts a long history of use in traditional Ayurvedic and Chinese medicine for its anti-inflammatory and antioxidant properties [69]. Extensive research explores CU’s potential therapeutic effects in various diseases, including cancer [70]. Our investigation demonstrates that this natural compound, when combined with DCA, significantly enhances the generation of MGO and AGEs in Jurkat cells, ultimately leading to apoptosis induction. This novel application of CU as a potential adjuvant therapy for leukemia underscores the value of drug reuse in oncology. This strategy capitalizes on the established safety profiles of existing compounds, thereby expediting the development of novel cancer treatments while substantially reducing the time and resources required compared to de novo drug discovery efforts.

In line with our observations, other studies suggest a synergistic effect between CU and other anticancer agents. For example, Vemuri et al. [71] investigated the molecular mechanisms underlying CU’s potential to impede metastasis when combined with paclitaxel. Their findings revealed that CU, both independently and in combination with paclitaxel, effectively suppressed the expression of metastasis-associated factors, including vascular endothelial growth factor (VEGF), cyclin D1, and STAT3. This suppression resulted in the downregulation of the genes encoding these factors. Furthermore, CU treatment demonstrated a pro-apoptotic effect by upregulating the pro-apoptotic protein caspase-9. CU influence on MGO levels might be multifaceted. While it is a well-established byproduct of cellular processes, some studies suggest CU may inhibit the activity of the glyoxalase enzyme, which is responsible for MGO degradation [72]. This could potentially lead to an increase in MGO levels, which is consistent with our observation that CU treatment increases MGO levels.

Current evidence suggests a nuanced role for CU in MGO metabolism within the context of cancer therapy. While a direct effect on MGO production remains elusive, CU’s potential to modulate glyoxalase activity and its established synergy with other anticancer agents warrant further investigation. Elucidating the specific mechanisms underlying these interactions is crucial to determining whether selective drug combinations can elevate MGO levels, leading to targeted cancer cell death. Future research efforts should explore how CU interacts with metabolic pathways that regulate MGO and investigate the potential for combination therapies that contribute enhanced therapeutic benefits.

Drug repurposing offers a compelling strategy for oncology, characterized by expedited development timelines, reduced financial investments, and established safety profiles. Our study exemplifies this potential by repurposing DS, CU, and DCA for leukemia treatment. Capitalizing on the existing pharmacological properties and safety data of these compounds, our work underscores a cost-effective and time-efficient approach to novel cancer therapeutic development. The encouraging results of our research not only highlight the efficacy of these drugs in sensitizing cancer cells to therapies but also pave the way for further exploration of drug repurposing strategies. By optimizing these strategies, we can potentially achieve significant improvements in cancer treatment outcomes.

## 5. Conclusions

In conclusion, our study establishes the efficacy of DCA in sensitizing Jurkat cells to DS- and CU-mediated cytotoxicity. This enhanced susceptibility is associated with the induction of TPI isoforms, a concomitant increase in MGO and AGEs, and the activation of the apoptotic mechanism. These findings provide valuable insights into the metabolic and apoptotic signaling cascades implicated in leukemia therapy. Furthermore, they suggest promising avenues for the development of targeted therapeutics that exploit these mechanisms. Future efforts should be directed toward evaluating the clinical translatability of these drug combinations in a broader spectrum of malignancies and towards a more comprehensive understanding of the underlying molecular pathways.

The use of only one cell line restricts the generalizability of our results, highlighting the need for further studies to validate these findings across a broader spectrum of T-ALL cell lines and in vivo models. Additionally, the long-term effects and potential resistance mechanisms related to DCA, DS, and CU treatments remain to be investigated. Therefore, further studies must be performed on evaluating the clinical translatability of these drug combinations in diverse malignancies and on gaining a more comprehensive understanding of the underlying molecular pathways.

## Figures and Tables

**Figure 1 biomolecules-14-01295-f001:**
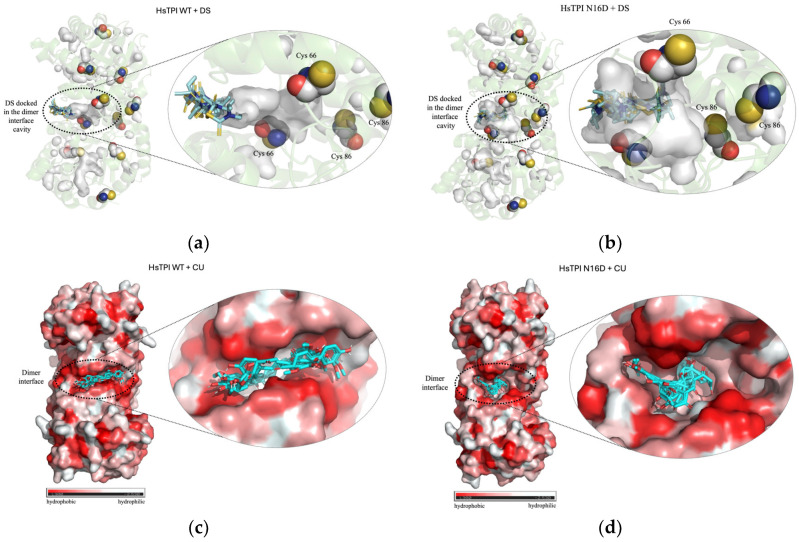
Molecular docking analysis of n-dTPI and dTPI crystallographic structures. Conformational representations of DS docked at the interface of n-dTPI (**a**) and dTPI (**b**) are shown. The proximity of Cys residues near the interface is highlighted, suggesting potential interaction sites for DS. Conformational representations of CU docked at the interface of n-dTPI (**c**) and dTPI (**d**) are shown; the hydrophobic surfaces of proteins are highlighted in red. A deeper penetration of both DS and CU into the interface of the deamidated enzyme (dTPI) compared to the non-deamidated form (n-dTPI) is observed. Figures modeled with PyMOL version 2.5.0 (Schrödinger Inc., New York, NY, USA).

**Figure 2 biomolecules-14-01295-f002:**
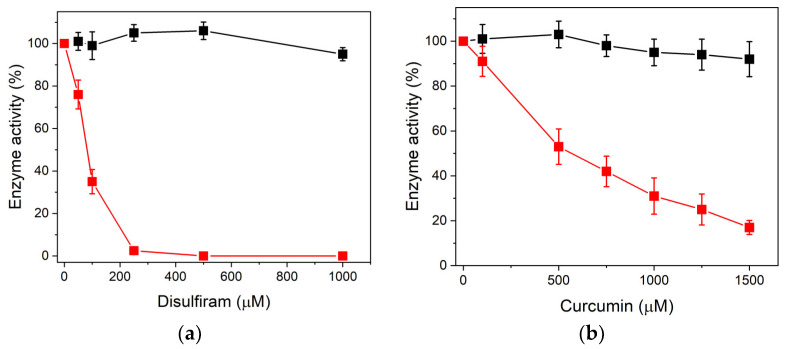
Inactivation assays of recombinant n-dTPI and dTPI enzymes. Both enzymes (0.2 mg/mL) were incubated for 2 h at 37 °C with gradually increasing concentrations of the respective compounds: (**a**) DS (0 to 1000 μM) and (**b**) CU (0 to 1500 μM). Following incubation, aliquots were taken for enzyme activity measurement as described in the Methods section. The enzymatic activity was normalized, and 100% corresponds to the assays in the absence of the compound. Filled black squares represent n-dTPI activity, while filled red squares represent dTPI activity. The results represent the average of three independent experiments, with error bars indicating the variation observed in the experiments.

**Figure 3 biomolecules-14-01295-f003:**
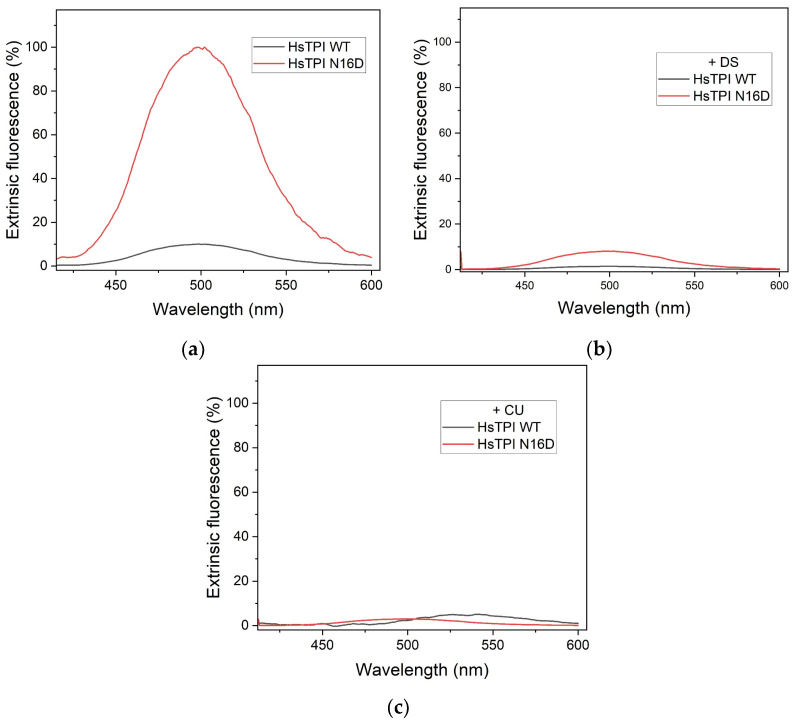
Extrinsic fluorescence spectra of TPIs after incubation with DS and CU. Enzymes (0.2 mg/mL) were incubated without or with 250 μM DS and 1500 μM CU for 2 h at 37 °C. Following incubation, excess compounds were removed, and extrinsic fluorescence was measured in the presence of 100 μM ANS with excitation at 395 nm. (**a**) shows the spectra in the absence of any compounds (control). (**b**,**c**) show the spectra after incubation with DS and CU, respectively. n-dTPI (black line) and dTPI (red line). The results represent the mean of three independent experiments.

**Figure 4 biomolecules-14-01295-f004:**
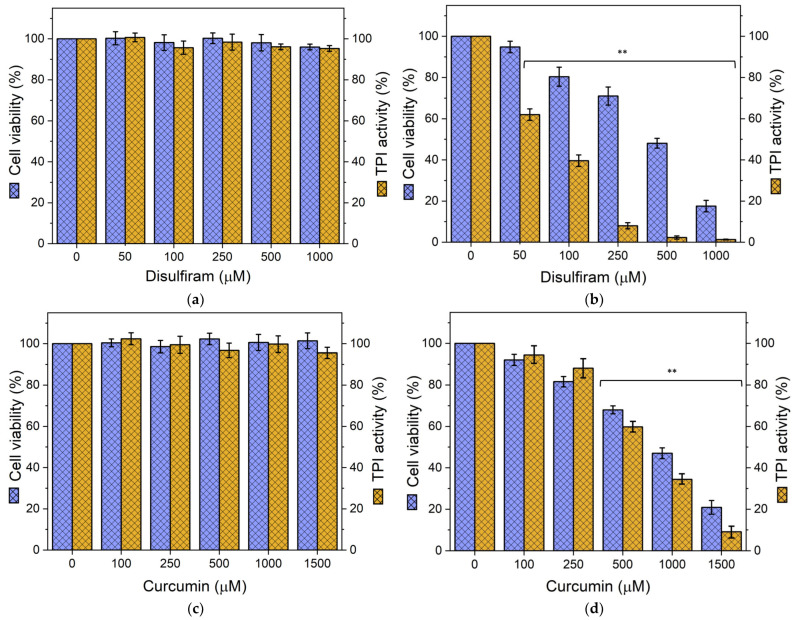
Effects of DS and CU on cell viability and TPI activity in normal T-cell lymphocytes and Jurkat cells. Cells (1 × 10^5^ per well) were incubated with increasing concentrations of DS and CU. Following incubation, cell viability was assessed using MTT assays and endogenous TPI activity was determined by enzymatic activity assays. (**a**,**c**) normal T lymphocytes, (**b**,**d**) Jurkat cells. Results are expressed as percentages relative to the untreated control set to 100%. The results represent the average of three independent experiments, with error bars indicating the variation observed in the experiments. Statistical differences were analyzed using one-way ANOVA with Tukey’s post-hoc test, with a significance level set at *p* = 0.01 (**).

**Figure 5 biomolecules-14-01295-f005:**
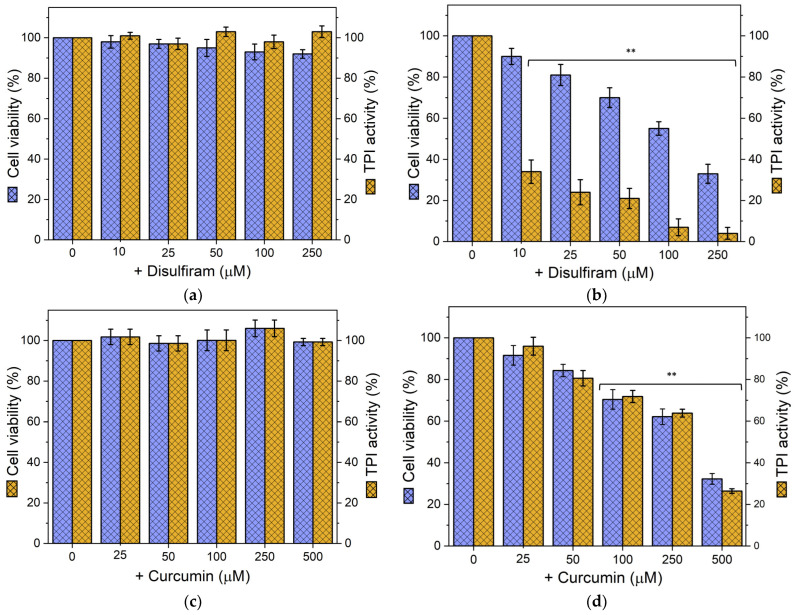
Effects of combined DCA and CU treatment on cell viability and TPI activity in normal T lymphocytes and Jurkat cells. Cells (1 × 10^5^ per well) were pre-treated with 12 mM DCA for 24 h at 37 °C, followed by exposure to increasing concentrations of DS or CU for an additional 24 h. MTT and enzyme activity assays were then performed to assess cell viability and endogenous TPI activity, respectively. (**a**,**c**) normal T lymphocytes and (**b**,**d**) Jurkat cells. Results are expressed as percentages relative to the untreated control group (set to 100%). The results represent the average of three independent experiments, with error bars indicating the variation observed in the experiments. Statistical differences were analyzed using one-way ANOVA with Tukey’s post-hoc test, with a significance level set at *p* = 0.01 (**).

**Figure 6 biomolecules-14-01295-f006:**
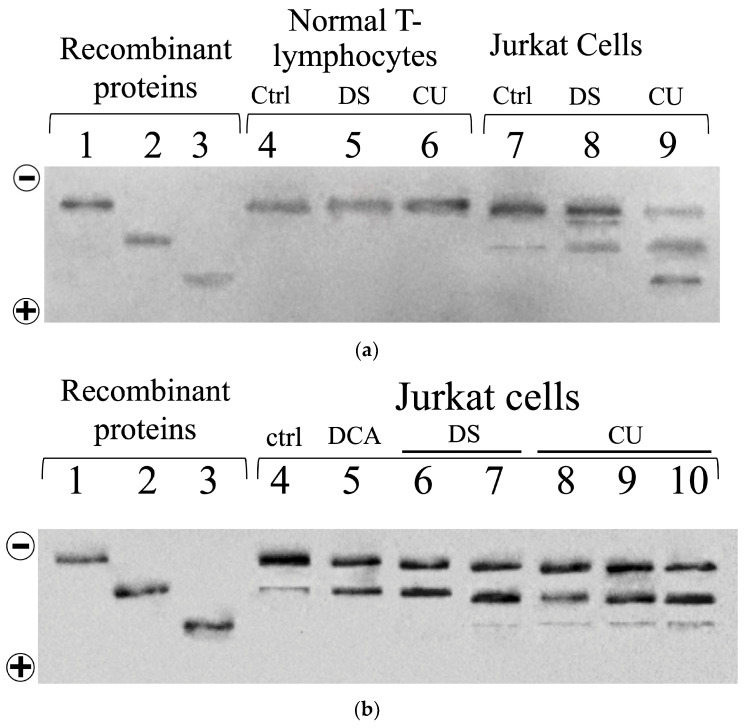
Western blot analysis of TPI isoforms in normal T lymphocytes and Jurkat cells. In (**a**) and (**b**), lanes 1, 2, and 3, each loaded with 1 μg of protein, serve as migration standards for recombinant n-dTPI, dTPI, and ddTPI, respectively. Lanes 4–6 in (**a**) contain total protein extracts from normal T lymphocytes, while lanes 7–9 in (**a**) and lanes 4–10 in (**b**) correspond to protein extracts from Jurkat cells, with 100 μg of protein loaded per lane. For normal T lymphocytes, lane 4 is the control (untreated), lane 5 is treated with 250 μM DS, and lane 6 is treated with 1500 μM CU. In Jurkat cells, lanes 7, 8, and 9 in (**a**) represent the control (untreated), treatment with 250 μM DS, and treatment with 1500 μM CU, respectively. In (**b**), lane 4 is the control condition (untreated), while lanes 6 and 7 were treated with 100 and 250 μM DS, respectively. Finally, lanes 8, 9, and 10 were treated with 500, 1000, and 1500 μM of CU, respectively. The positive and negative poles of the gel are indicated on the left side of each panel. Full-length (uncropped) blots for panels (**a**,**b**) are presented in Appendix A.

**Figure 7 biomolecules-14-01295-f007:**
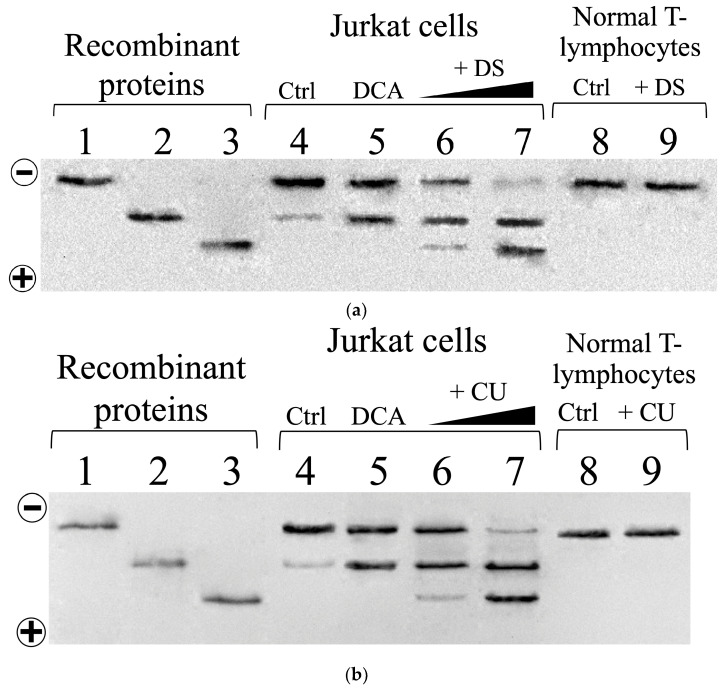
Western blot analysis of TPI isoforms in normal T-cell lymphocytes and Jurkat cells with combined DCA and CU treatment. In both panels, lanes 1–3 represent recombinant n-dTPI, dTPI, and ddTPI enzymes loaded at 1 μg protein per lane, serving as migration standards. In both panels, lanes 4–9 with 100 μg of protein loaded per lane. In a and b, lane 4 shows control Jurkat cells (untreated); lane 5 shows 12 mM DCA. In (**a**) lane 6 and 7 are Jurkat cells pretreated with 12 mM DCA followed by incubation with 100 and 250 μM DS, respectively. In (**b**) lane 6 and 7 are Jurkat cells pretreated with 12 mM DCA followed by incubation with 1000 μM and 1500 μM CU, respectively. Finally, in both, a and b, lanes 8–9 show normal T lymphocytes pretreated with 12 mM DCA followed by incubation with 250 μM DS (**a**) or 1500 μM CU (**b**). The polarity of the gel is indicated on the left side of the panel. Full-length blots (uncropped blots) are shown in Appendix A.

**Figure 8 biomolecules-14-01295-f008:**
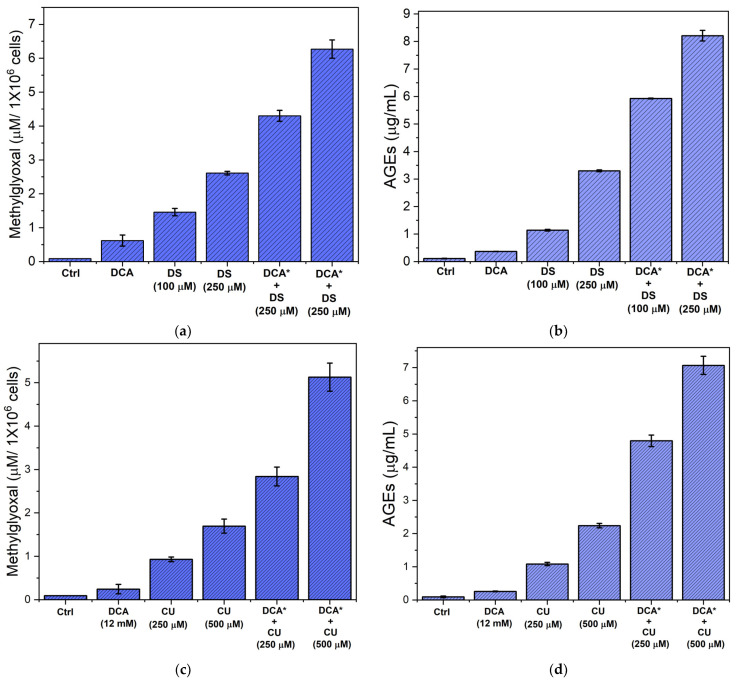
MGO and AGEs levels in Jurkat cells following DCA, DS, and CU treatment. Quantification of MGO (**a**,**c**) and AGEs (**b**,**d**) in Jurkat cells following treatment with DCA, DS, CU, or their combination (*). As observed, increasing concentrations of DS and CU lead to a dose-dependent rise in both MGO and AGEs. Notably, pretreatment with DCA before DS or CU administration results in a further significant increase in MGO and AGEs production. The results represent the average of three independent experiments, with error bars indicating the variation observed in the experiments.

**Figure 9 biomolecules-14-01295-f009:**
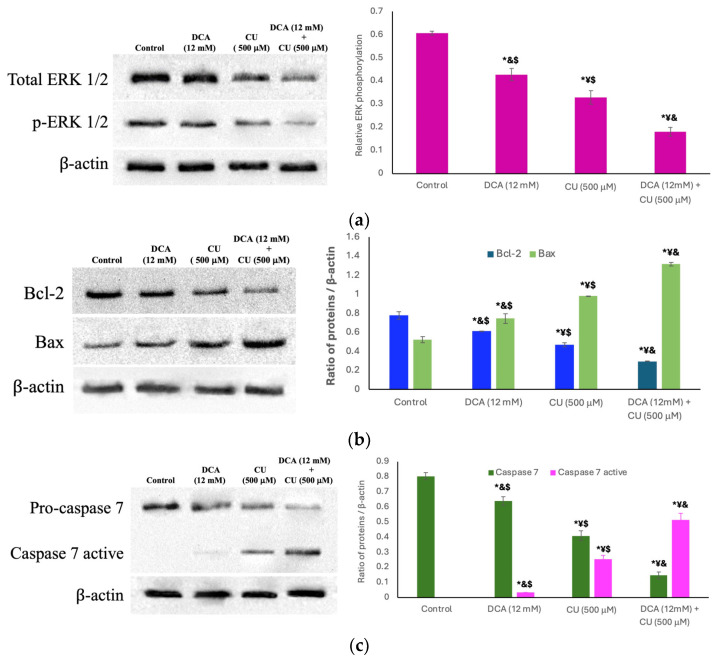
Western blot analysis of apoptosis-related proteins in Jurkat cells. (**a**) Expression of ERK1/2 and its phosphorylated form following treatment with DCA, CU, or their combination. The corresponding bar graphs quantify the decrease in total and phosphorylated ERK1/2 levels relative to the control. (**b**) Expression of Bcl-2 and Bax proteins under treatment with DCA, CU, or their combination. The bar graphs represent the relative expression levels of these proteins. (**c**) Procaspase-7 and cleaved caspase-7 levels following treatment with DCA, CU, or their combination. The bar graphs quantify the relative abundance of procaspase-7 and cleaved caspase-7 compared to the control. β-Actin was used as a loading control for all Western blots. Each lane was loaded with 100 μg of total protein extract. Statistical analysis was performed using a one-way ANOVA followed by the Tukey-Kramer test. Significance levels are indicated as follows: * *p* ≤ 0.01 compared to the control, ¥ *p* ≤ 0.01 compared to 12 mM DCA treatment, & *p* ≤ 0.01 compared to 0.5 mM CU treatment, $ *p* ≤ 0.01 compared to 12 mM DCA + 0.5 mM CU treatment. Full-length blots (uncropped blots) are shown in Appendix A for Total ERK 1/2, pERK, and β-Actin, respectively; Appendix A for Bcl2, Bax, and β-Actin, respectively, and Appendix A for Procaspase-7, cleaved caspase-7, and β-Actin, respectively.

**Figure 10 biomolecules-14-01295-f010:**
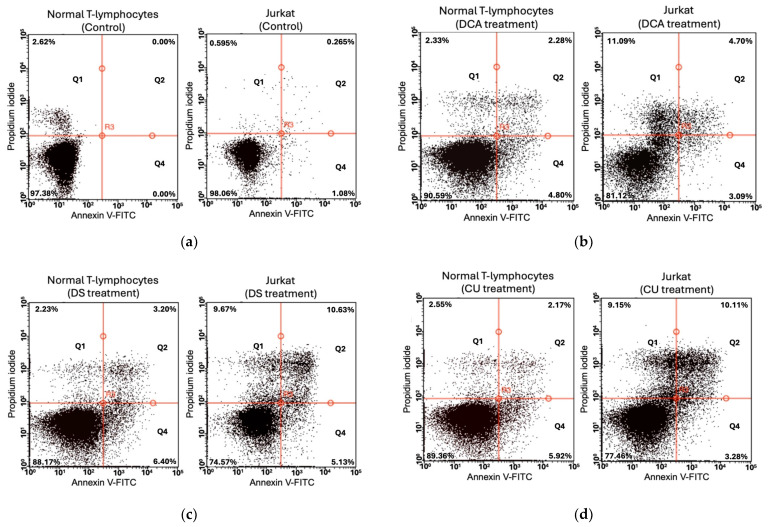
Detection of cell death in normal T lymphocytes and Jurkat cells after treatment with DCA, DS, CU, or their combinations (DCA + DS or DCA + CU). Cells were incubated with 12 mM DCA, 100 µM DS, or 250 µM CU for 24 h at 37 °C. For combination treatments, cells were first exposed to 12 mM DCA for 24 h, followed by an additional 24-h incubation with 100 µM DS or 250 µM CU. At the end of the incubation period, cells were washed, resuspended at a density of 1 × 10^6^/mL, and stained with Annexin V and Propidium Iodide, for flow cytometry analysis. (**a**) shows untreated control cells. (**b**–**d**) represent cells treated with DCA, DS, and CU, respectively. (**e**,**f**) display cells pretreated with DCA followed by DS and CU, respectively. Quadrants indicate distinct cell populations: Q1 (necrotic cells), Q2 (late apoptotic cells), and Q4 (early apoptotic cells). The data presented in the figure are representative of 100,000 cells analyzed across two independent experiments.

**Table 1 biomolecules-14-01295-t001:** Quantification of Cys residues in recombinant TPIs treated with DS or CU.

Enzyme	DS Treatment	^1^ Free Cys Residues/Subunit	^2^ Derivatized Cys Residues/Subunit
**n-dTPI**	-	5.2 ± 0.3	^3^ NA
	+	3.9 ± 0.2	~1
**dTPI**	-	4.8 ± 0.3	NA
	+	2.1 ± 0.3	~3
	**CU treatment**		
**n-dTPI**	-	4.8 ± 0.3	NA
	+	4.9 ± 0.4	0
**dTPI**	-	4.7 ± 0.2	NA
	+	4.9 ± 0.3	0

^1^ Quantification of Cys residues was performed under denatured conditions. ^2^ Derivatized Cys residues were determined by subtracting the control values from the experimental values. ^3^ Not applicable.

## Data Availability

Data are contained within the article.

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
