# Peer review of "Selective Inhibition of Deamidated Triosephosphate Isomerase by Disulfiram, Curcumin, and Sodium Dichloroacetate: Synergistic Therapeutic Strategies for T-Cell Acute Lymphoblastic Leukemia in Jurkat Cells"

_biomolecules, 2024, doi:10.3390/biom14101295_

Round 1
Reviewer 1 Report
Comments and Suggestions for Authors
The authors analyze the impacts of disulfiram and curcumin in combination with dichloroacetic acid on dTPI in T-ALL cells. I believe the authors do a thorough job of exploring these effects and utilize impactful controls, normal cells and non-deamidated enzyme effects, to demonstrate a clear impact of these compounds. The analysis utilizes many approaches to cover the aspects of the story appropriately, demonstrating the potential of the dTPI target and use of the compounds. There are some things I suggest clarifying and adding to help strengthen the manuscript as outlined in depth below. I recommend to accept with minor revisions.
Abstract – the first sentence is about drug resistance but there is no real analysis of resistance in the manuscript. The cells used are not relapsed T-ALL or made to be resistant. I think it should be edited since it makes the reader think there is a resistance focus.
Introduction – it is mentioned in the discussion that DCA has been developed for metabolic disorders. It would be beneficial to briefly mention it is approved for use for these conditions in the introduction to clarify to the reader why it was selected before diving into the rest of the paper.
Methods – for the enzyme activity, cell viability assays, western blots, apoptosis, etc. (any experiment compared to “untreated”), it is mentioned that results are a percentage compared to untreated cells or “absence of the drug” (line 206-207 section 2.4). However, it says the compounds were dissolved in 100% ethanol (line 195 section 2.4) for the enzyme activity assay. Was ethanol used as a vehicle for all assays? If so, the “untreated” control should have been equal percentages of ethanol. Can the authors confirm that the control is a vehicle control with equal ethanol and not just untreated for all assays? Otherwise, the control is not taking into consideration the potential effects of ethanol. Ethanol could induce death and other cell effects. All treatments should have the same percentage of ethanol.
Results
Lines 430-431 “compared to the significantly higher values observed for the deamidated form”. The word significant implies some use of statistics to determine it so, but I do not think there was a type of statistical analysis used, since it is difficult to do so with docking analysis, so using significantly is misleading.
3.1.1, table 1 - Why was the Cys residue experiment only done with DS and not curcumin? I think the authors should analyze these effects for curcumin treatment as well.
Figure 4 – the y-axis could be more specific. I realize it is for two things but maybe: percent of untreated (or vehicle) control.
Figure 4 - for a, b and c there is a concentration label missing on the x-axis, I believe it would be 50 uM for a and b, and 100 for c.
Figure 4 & 5 - The statistics are a little confusingly explained for 4 and 5 and should be clarified in the caption. Is it that all those bars within the bracket are statistically different (**) from the control of their respective viability or enzyme activity untreated/vehicle control? In the caption it just says “statistical difference between groups were analyzed” so it is not clear.
I think it would be beneficial for the authors to quantify the IC50 for the T-ALL/Jukrat cells for each drug since the data is already available to do so (viability data from figure 4 can be used). This would allow for an easier comparison in their viability reduction effectiveness.
Caption for figure 4 and 5 says “cancer cells” best to include T-ALL or Jurkrat cancer cells for specificity (line 576, line 622).
Figure 6, section 3.2.3 – I am a little perplexed by the increase of dTPI and ddTPI with CU treatment. It is demonstrated earlier that there is decreasing enzymatic activity and targeting of dTPI with CU. I understand that is activity reduction which is also shown thoroughly with the MGO and AGE experiment. However, cancer cells upregulate these forms, which is shown in comparison of Jukrat cells to normal T-lymphocytes, so that appears to be a metabolic switch that is undesirable. There is an explanation in lines 700-704 about how the increased expression with DCA and CU treatment could make higher drug-sensitive variants. However, with the previous experiments the authors are saying CU targets the activity so is the use of CU being suggested to target the activity or induce expression for other drugs to target its activity?
Figure 7 – there is a discrepancy between what is lane 8. In the picture it says T-lymphocyte CTRL but in the description it says that lane was “treated with 12 mM DCA”. Either way one of those, the control with no treatment and the control of just DCA is missing. Please clarify.
Discussion/introduction - Is deamidated TPI more active in glycolysis than non-deamidated? Evidence of upregulation of deamidated TPI in cancer cells is mentioned and there are comments in the introduction lines 75-79 on how this PTM can impact activity and stability but it is not mentioned whether it increases or decreases activity. Just want to clarify if this increase in dTPI may be part of why glycolysis activity is increased in cancer cells. It would be helpful to explain how it impacts its activity (or if it does not) in the discussion or introduction.
Conclusion – line 1035 “the dose-dependent efficacy of DCA” this is confusing because only one dose/concentration of DCA was used per experiment when combined with CU.
Why is there no comment about DS in the conclusion? It seems the authors have abandoned DS as we get deeper into the discussion into conclusion. Is there a reason for this?
Comments on the Quality of English LanguageOverall, the English quality was good. Some minor grammatical errors.
Author Response
Dear Reviewers,
We sincerely appreciate the time and effort you have dedicated to reviewing our manuscript titled “Deamidated triosephosphate isomerase as a selective target for T-ALL
therapy: Synergistic inhibition by dichloroacetic acid and curcumin.” (biomolecules-3202107). Your insightful comments and constructive feedback have been invaluable in improving the quality and clarity of our work.
We have carefully considered each of your comments and suggestions. Below, we provide detailed, point-by-point responses to all the issues raised. Changes made to the manuscript have been highlighted in red for your convenience.
Response to reviewer #1
Comment 1. Abstract – the first sentence is about drug resistance but there is no real analysis of resistance in the manuscript. The cells used are not relapsed T-ALL or made to be resistant. I think it should be edited since it makes the reader think there is a resistance focus.
Repply:
Thank you for your valuable feedback. We have revised the first sentence of the abstract to remove references to drug resistance, as the manuscript does not focus on this aspect. The updated sentence now accurately reflects the study's emphasis on the therapeutic implications without suggesting an analysis of resistance.
Comment 2. Introduction – it is mentioned in the discussion that DCA has been developed for metabolic disorders. It would be beneficial to briefly mention it is approved for use for these conditions in the introduction to clarify to the reader why it was selected before diving into the rest of the paper.
Repply:
Thank you for your suggestion. We have added a brief mention in the introduction highlighting that sodium dichloroacetate (DCA) is a molecule used in metabolic disorders among others. This clarification provides context for its selection in our study and enhances the reader's understanding before we delve into the subsequent sections of the paper.
Comment 3. Methods – for the enzyme activity, cell viability assays, western blots, apoptosis, etc. (any experiment compared to “untreated”), it is mentioned that results are a percentage compared to untreated cells or “absence of the drug” (line 206-207 section 2.4). However, it says the compounds were dissolved in 100% ethanol (line 195 section 2.4) for the enzyme activity assay. Was ethanol used as a vehicle for all assays? If so, the “untreated” control should have been equal percentages of ethanol. Can the authors confirm that the control is a vehicle control with equal ethanol and not just untreated for all assays? Otherwise, the control is not taking into consideration the potential effects of ethanol. Ethanol could induce death and other cell effects. All treatments should have the same percentage of ethanol.
Repply:
Thank you for your important observation regarding the control conditions. We confirm that ethanol was used as a vehicle for all assays (at 1.5% final concentration). To address your concern, we have clarified in the Methods section that the "untreated" controls for all experiments were matched with equal percentages of ethanol to ensure accurate comparisons. This modification ensures that any potential effects of ethanol on cell viability, enzyme activity, and apoptosis are accounted for.
Comment 4. Results – Lines 430-431 “compared to the significantly higher values observed for the deamidated form”. The word significant implies some use of statistics to determine it so, but I do not think there was a type of statistical analysis used, since it is difficult to do so with docking analysis, so using significantly is misleading.
Repply:
Thank you for pointing out this issue regarding the terminology used in the Results section. We acknowledge that the term "significantly" implies statistical analysis, which is not applicable to the docking analysis presented. We have revised the wording to remove the term "significantly" and replaced it with more appropriate language that accurately reflects the findings without implying statistical significance.
Comment 5. 3.1.1, table 1 - Why was the Cys residue experiment only done with DS and not curcumin? I think the authors should analyze these effects for curcumin treatment as well.
Repply:
Thank you for your insightful comment regarding the Cys residue experiment. We recognize the importance of examining the effects of curcumin treatment alongside those of DS. To address this, we have conducted additional experiments to analyze the impact of curcumin on Cys residues. The results of these analyses have been included in the revised Table 1 and discussed in the Results section to provide a comprehensive understanding of the effects of both compounds.
Comment 6. Figure 4 – the y-axis could be more specific. I realize it is for two things but maybe: percent of untreated (or vehicle) control.
Repply:
Thank you for your valuable suggestion regarding the y-axis labeling in Figure 4. We have revised the axis labels in Figure 4, as well as in the subsequent figures, to enhance clarity. The y-axes now include specific labels that provide clearer context for the data presented. Additionally, we have clarified in both the main text and the corresponding figure caption that the percentage of control corresponds to the used vehicle.
Comment 7. Figure 4 - for a, b and c there is a concentration label missing on the x-axis, I believe it would be 50 uM for a and b, and 100 for c.
Repply:
You are right, we have added the appropriate concentration labels, indicating 50 µM for panels a and b, and 100 µM for panel c.
Comment 8. Figure 4 & 5 - The statistics are a little confusingly explained for 4 and 5 and should be clarified in the caption. Is it that all those bars within the bracket are statistically different (**) from the control of their respective viability or enzyme activity untreated/vehicle control? In the caption it just says “statistical difference between groups were analyzed” so it is not clear.
Repply:
Thank you for your feedback regarding the statistical analysis presented in Figures 4 and 5. We recognize that the explanation in the figure captions was not sufficiently clear. We have revised the captions to explicitly state that all bars within the brackets are statistically different (**) from the respective untreated or vehicle control for both viability and enzyme activity.
Comment 9. I think it would be beneficial for the authors to quantify the IC50 for the T-ALL/Jukrat cells for each drug since the data is already available to do so (viability data from figure 4 can be used). This would allow for an easier comparison in their viability reduction effectiveness.
Repply:
Thank you for your valuable suggestion regarding the quantification of IC50 values for the T-ALL/Jukrat cells for each drug. We agree that providing these calculations would enhance the comparison of the drugs' effectiveness in reducing cell viability. We have now quantified the IC50 values using the viability data from corresponding Figures and included the results in the revised manuscript.
Comment 10. Caption for figure 4 and 5 says “cancer cells” best to include T-ALL or Jurkrat cancer cells for specificity (line 576, line 622).
Repply:
Thank you, we have reviewed the entire manuscript and normalized for control cells “normal T-lymphocytes” and for cancer cells “Jurkat cells”. This information has been integrated into the relevant sections to improve clarity and consistency throughout the text.
Comment 11. Figure 6, section 3.2.3 – I am a little perplexed by the increase of dTPI and ddTPI with CU treatment. It is demonstrated earlier that there is decreasing enzymatic activity and targeting of dTPI with CU. I understand that is activity reduction which is also shown thoroughly with the MGO and AGE experiment. However, cancer cells upregulate these forms, which is shown in comparison of Jukrat cells to normal T-lymphocytes, so that appears to be a metabolic switch that is undesirable. There is an explanation in lines 700-704 about how the increased expression with DCA and CU treatment could make higher drug-sensitive variants. However, with the previous experiments the authors are saying CU targets the activity so is the use of CU being suggested to target the activity or induce expression for other drugs to target its activity?
Repply:
Thank you for your thought-provoking question regarding the increase in dTPI and ddTPI with treatments of curcumin (CU), disulfiram (DS), and sodium dichloroacetate (DCA), particularly in light of the observed decrease in enzymatic activity. You raise an important point about the apparent metabolic switch in cancer cells, where upregulation of these isoforms may occur despite reduced enzymatic activity.
We believe it is essential to consider both perspectives. Our data on endogenous TPI activity indicate that exposure of Jurkat cells to CU results in decreased enzymatic activity. However, analysis of native gels reveals a relative increase in the deamidated isoforms of TPI across all treatment conditions, suggesting a potential mechanism of accumulation for these isoforms. This enrichment of "vulnerable" TPI isoforms may create a vicious cycle, enabling Jurkat cells to continually produce these susceptible forms. This finding is consistent with the results reported by Enriquez-Flores et al. (2022) [1], which demonstrated an increase in acidic isoforms in a cellular model of breast cancer. Additionally, other studies have confirmed the presence and accumulation of acidic isoforms of TPI in lymphoblasts stimulated with mitogens, attributing the multiple variants observed in native gels (nPAGE) to deamidation [2].
Thus, while DS, CU appears to target TPI activity directly, the concomitant increase in dTPI and ddTPI expression may indicate a strategy for enhancing sensitivity to additional therapeutic agents. We appreciate your insightful query, which has prompted us to clarify this complex interaction in the discussion.
Comment 12. Figure 7 – there is a discrepancy between what is lane 8. In the picture it says T-lymphocyte CTRL but in the description it says that lane was “treated with 12 mM DCA”. Either way one of those, the control with no treatment and the control of just DCA is missing. Please clarify.
Repply:
Thank you for pointing out the discrepancy regarding lane 8 in Figure 7. We have revised both the figure and its legend for accuracy. Specifically, we clarified whether lane 8 represents the untreated T-lymphocyte control or cells treated with 12 mM DCA, ensuring that all relevant controls are now clearly indicated.
Additionally, as requested, we have enhanced the quality of the native gel images and incorporated further studies using disulfiram (DS) to complement the experimental data. These revisions provide a more comprehensive view of the treatment effects and ensure clearer visual representation in the figures. We appreciate your constructive feedback, which has allowed us to improve the presentation of the data.
Comment 13. Discussion/introduction - Is deamidated TPI more active in glycolysis than non-deamidated? Evidence of upregulation of deamidated TPI in cancer cells is mentioned and there are comments in the introduction lines 75-79 on how this PTM can impact activity and stability but it is not mentioned whether it increases or decreases activity. Just want to clarify if this increase in dTPI may be part of why glycolysis activity is increased in cancer cells. It would be helpful to explain how it impacts its activity (or if it does not) in the discussion or introduction.
Repply:
Thank you for your thoughtful inquiry regarding the activity of dTPI in glycolysis compared to n-dTPI. We acknowledge the necessity of clarifying the implications of this post-translational modification on TPI activity, particularly considering its upregulation in cancer cells.
As noted, the activities of endogenous cellular TPI have not yet been thoroughly evaluated, meaning we currently lack direct comparisons between n-d TPI and dTPI. Isolating and enriching the isoforms of endogenous cellular TPI is challenging. However, previous studies of recombinant enzymes have shown that deamidated TPI has lower enzymatic activity than its non-deamidated counterpart [3]. Moreover, research involving a bacterial cell model lacking TPI demonstrated significant differences in enzymatic activities when complemented with both non-deamidated and deamidated TPI [4].
This implies that a similar scenario may occur in the cellular context. Given that TPI catalysis can be almost limited by the diffusion of its substrates, even with reduced activity, TPI can still provide the necessary substrates for glycolysis. However, once TPI is deamidated, its structure undergoes significant alterations. Consequently, the progressive accumulation of dTPI and its enrichment within the cellular environment may render it more susceptible to attack by small molecular weight compounds.
We have addressed these points in the corresponding sections to enhance the reader's understanding of the effects of deamidated TPI accumulation and the vulnerabilities associated with it.
Comment 14. Conclusion – line 1035 “the dose-dependent efficacy of DCA” this is confusing because only one dose/concentration of DCA was used per experiment when combined with CU.
Repply:
Thank you for your comment. We recognize that this wording may cause confusion, as only one concentration of DCA was used in each experiment when combined with curcumin (CU). We have amended the conclusion to clarify that the experiments focused on a single dose of DCA in conjunction with DS and CU, rather than involving a broader assessment of dose-dependent effects.
Comment 15. Why is there no comment about DS in the conclusion? It seems the authors have abandoned DS as we get deeper into the discussion into conclusion. Is there a reason for this?
Repply:
Thank you for your valuable feedback regarding the lack of mention of DS in the conclusion. We appreciate your observation that it may appear as though DS has been overlooked as we progress through the discussion and into the conclusion.
We want to clarify that experiments involving DS were conducted throughout the manuscript to determine its efficacy both alone and in combination with sodium dichloroacetate (DCA). However, it seems that we did not adequately emphasize its significance in the conclusion.
We revised all manuscript to include the findings related to DS, highlighting its role in the experiments and its potential therapeutic implications.
Comment 16. Comments on the Quality of English Language. Overall, the English quality was good. Some minor grammatical errors.
Repply:
Thank you for your positive feedback on the quality of the English language in our manuscript. We are pleased to hear that you found the overall quality to be good. We appreciate your noting the presence of minor grammatical errors, and we will conduct a thorough review to address these issues and enhance the clarity and readability of the text. Your attention to detail is greatly valued and will contribute to the overall quality of our work.
References.
- Enríquez-Flores, S., Flores-López, L. A., De la Mora-De la Mora, I., García-Torres, I., Gracia-Mora, I., Gutiérrez-Castrellón, P., Fernández-Lainez, C., Martínez-Pérez, Y., Olaya-Vargas, A., de Vos, P., & López-Velázquez, G. (2022). Naturally occurring deamidated triosephosphate isomerase is a promising target for cell-selective therapy in cancer. Scientific reports, 12(1), 4028. https://doi.org/10.1038/s41598-022-08051-0.).
- Yuan, P. M., Talent, J. M., & Gracy, R. W. (1981). Molecular basis for the accumulation of acidic isozymes of triosephosphate isomerase on aging. Mechanisms of ageing and development, 17(2), 151–162. https://doi.org/10.1016/0047-6374(81)90081-6.).
- De la Mora-de la Mora, I., Torres-Larios, A., Enríquez-Flores, S., Méndez, S. T., Castillo-Villanueva, A., Gómez Manzo, S., López-Velázquez, G., Marcial-Quino, J., Torres-Arroyo, A., García-Torres, I., Reyes-Vivas, H., & Oria-Hernández, J. (2015). Structural effects of protein aging: terminal marking by deamidation in human triosephosphate isomerase. PloS one, 10(4), e0123379. https://doi.org/10.1371/journal.pone.0123379.
- Enríquez-Flores, S., Flores-López, L. A., García-Torres, I., de la Mora-de la Mora, I., Cabrera, N., Gutiérrez-Castrellón, P., Martínez-Pérez, Y., & López-Velázquez, G. (2020). Deamidated Human Triosephosphate Isomerase is a Promising Druggable Target. Biomolecules, 10(7), 1050. https://doi.org/10.3390/biom10071050.
Reviewer 2 Report
Comments and Suggestions for Authors
This study by Florez-Lopez and colleagues is a thorough biochemical characterization of the effects of 3 different compounds on the deamidated triosephosphate isomerase (DTI) enzyme. The authors tested the compounds as single-agents or in combination on healthy T lymphocytes or a T-ALL cell line (Jurkat). Given that DTI is a key enzyme for glycolysis, and considering that T-ALL cells rely on glycolytic metabolism, the authors investigate the biochemical and biological effects of DTI targeting. They first offer a molecular docking analysis of two compounds (CU and DS) on dTPI and n-dTPI enzymes, further validating their in silico predictions by in vitro enzymatic assays. They further evaluate the effects of both compounds in cell culture, showing a specificity for T-ALL and not for healthy T cells. Then they propose a combinatorial treatment by pre-treating cells with DCA and then administering CU. They show that DCA sensitizes Jurkat cells to CU-mediated cytotoxicity, mechanistically demonstrating that the combinatorial treatment causes an increase in MGO and AGE, leading to increased levels of apoptosis. Overall, the authors propose the DCA+CU combination as a novel / repurposed drug treatment for T-ALL.
The study is well-conducted and written, although there are a few syntactical errors throughout the text. Although potentially interesting from a clinical perspective, the study only considers 1 single cell line and lacks an in vivo characterization of treatment efficacy and safety.
The authors should keep in mind that the Jurkat cell line is a very limited model to describe T-ALL, which is a complex disease with multiple subtypes (some of them relying more on fatty acid metabolism than on glycolysis). In order to claim a more general effect on T-ALL, the authors should at least analyze a panel of commercially available cell lines (RPMI-8402, CUTLL1, DND41, to name a few). This narrow model, which only focuses on a single cell line, in my opinion is the main limitation of the study.
I also include additional points that should be considered to improve the impact of the results of this paper:
1) In figure 2, the authors show that both DS and CU have a strong inactivating function for dTPI enzymes while they don’t show any effect on the n-dTPI enzymes. Given the isoform specificity of the findings, have they tested (or has it been reported by others) that cancer cells have a higher proportion of dTPI than n-dTPI? I think an answer is already contained in the control lanes of figures 6 and 7, but it should be better underlined (maybe adding a quantification plot) and brought up in the introduction.
2) In a series of in vitro assays (figures 2 and 3), the authors show that both DS and CU efficiently inactivate dTPI enzymes. Since both drugs are predicted to bind the same pockets, have they tried a combinatorial administration of DS + CU? In case the inactivation turns out to be less efficient than the single drug, this would further prove a competitive binding for the same pockets.
3) Although the authors convincingly demonstrate the selectivity of DS and CU on Jurkat cell viability, to find an optimal dose of treatment for these compounds on Jurkat, an IC-50 curve should be provided.
4) In figure 5 the authors show the effects of combination treatments. They pre-treated Jurkat cells (or T lymphocytes) with DCA and then administered CU. It is not clear to me why they decided to only test CU for combinatorial treatments after showing (figure 4) that DS has a much higher potency without notable toxicity....
5) In figure 10 the authors convincingly show the apoptotic effects of DCA and/or CU administration to Jurkat cells. Although it has been previously shown that both compounds don’t alter the cell viability and TPI isoform representation in normal cells, for consistency they could include the same flow cytometry plots on T lymphocytes.
Comments on the Quality of English Language
There are some inaccuracies and syntactic errors, especially in the introduction section. To give a few examples, Line 56 : "relapse rates still need to be LOWER". Line 69: "TPI has not only GAINED significant attention".
Author Response
Dear Reviewers,
We sincerely appreciate the time and effort you have dedicated to reviewing our manuscript titled “Deamidated triosephosphate isomerase as a selective target for T-ALL
therapy: Synergistic inhibition by dichloroacetic acid and curcumin.” (biomolecules-3202107). Your insightful comments and constructive feedback have been invaluable in improving the quality and clarity of our work.
We have carefully considered each of your comments and suggestions. Below, we provide detailed, point-by-point responses to all the issues raised. Changes made to the manuscript have been highlighted in red for your convenience.
Response to reviewer #2
Comment 1. The authors should keep in mind that the Jurkat cell line is a very limited model to describe T-ALL, which is a complex disease with multiple subtypes (some of them relying more on fatty acid metabolism than on glycolysis). In order to claim a more general effect on T-ALL, the authors should at least analyze a panel of commercially available cell lines (RPMI-8402, CUTLL1, DND41, to name a few). This narrow model, which only focuses on a single cell line, in my opinion is the main limitation of the study.
Repply:
Thank you for your insightful comments regarding the use of the Jurkat cell line as a model for T-ALL. We acknowledge that the Jurkat cell line represents a limited model for studying this complex disease, which encompasses various subtypes, some of which may rely more on fatty acid metabolism than on glycolysis.
In response to your feedback, we have made the necessary changes throughout the manuscript, including revising the title to clearly indicate the limitations of our study. We have been careful to limit our interpretations to reflect the scope of our findings accurately. Furthermore, we have expanded the discussion section to explicitly address the limitations of using a single cell line, acknowledging the need for further studies that include a panel of commercially available.
Comment 2. In figure 2, the authors show that both DS and CU have a strong inactivating function for dTPI enzymes while they don’t show any effect on the n-dTPI enzymes. Given the isoform specificity of the findings, have they tested (or has it been reported by others) that cancer cells have a higher proportion of dTPI than n-dTPI? I think an answer is already contained in the control lanes of figures 6 and 7, but it should be better underlined (maybe adding a quantification plot) and brought up in the introduction.
Repply:
Thank you for your thoughtful comment regarding the isoform specificity of our findings in Figures 6 and 7, particularly concerning the proportions of deamidated TPI (dTPI) and non-deamidated TPI (n-dTPI) in cancer cells.
To address your question, we have indeed estimated the relative proportions of the TPI isoforms (dTPI and ddTPI) using densitometry analysis. This information has been discussed in the results section of our manuscript, and we have included a supplementary table illustrating the densitometry results for clarity.
We appreciate your suggestion to emphasize this finding, and we will ensure that it is more prominently highlighted to better contextualize our results.
Comment 3. In a series of in vitro assays (figures 2 and 3), the authors show that both DS and CU efficiently inactivate dTPI enzymes. Since both drugs are predicted to bind the same pockets, have they tried a combinatorial administration of DS + CU? In case the inactivation turns out to be less efficient than the single drug, this would further prove a competitive binding for the same pockets.
Repply:
Thank you for your valuable suggestion regarding the potential combinatorial administration of DS and CU. In response to your inquiry, we conducted experiments where dTPI was first incubated with DS, followed by CU, and vice versa.
The results, which are included in the results section of the manuscript, indicated that the inactivation of dTPI was primarily driven by the first compound introduced. This suggests that the binding sites of DS and CU may differ or that the inactivation mechanisms vary depending on the order in which the compounds are introduced.
Your insightful comment has led to a deeper understanding of the interactions between these compounds and their respective effects on dTPI activity.
Comment 4. Although the authors convincingly demonstrate the selectivity of DS and CU on Jurkat cell viability, to find an optimal dose of treatment for these compounds on Jurkat, an IC-50 curve should be provided.
Repply:
Thank you for your constructive feedback regarding the evaluation of the optimal dosing of DS and CU on Jurkat cell viability. In response to your suggestion, we have calculated and included the IC50 values for both DS and CU in the revised manuscript.
These values provide a clearer understanding of the dose-dependent effects of these compounds on Jurkat cell viability and enhance the interpretation of our findings.
Comment 5. In figure 5 the authors show the effects of combination treatments. They pre-treated Jurkat cells (or T lymphocytes) with DCA and then administered CU. It is not clear to me why they decided to only test CU for combinatorial treatments after showing (figure 4) that DS has a much higher potency without notable toxicity....
Repply:
Thank you for your insightful comment. While we acknowledge that DS exhibits higher potency with lower toxicity, our primary goal was to explore the potential for enhancing the therapeutic efficacy of CU when used in conjunction with DCA. Given the complementary mechanisms of action reported for CU and DCA, we aimed to assess whether their combined effects could yield beneficial outcomes for T-ALL therapy.
In response to your concern, we have included studies with DS in our manuscript to provide a comprehensive analysis of its effects, despite our focus on CU.
Comment 6. In figure 10 the authors convincingly show the apoptotic effects of DCA and/or CU administration to Jurkat cells. Although it has been previously shown that both compounds don’t alter the cell viability and TPI isoform representation in normal cells, for consistency they could include the same flow cytometry plots on T lymphocytes.
Repply:
Thank you for your constructive suggestion. We agree that adding these data would enhance the consistency of our findings and provide a clearer comparison between the effects of DCA and CU on both Jurkat and normal T-lymphocytes.
In response to your comment, we have included the relevant flow cytometry plots for normal T-lymphocytes in the revised version of the manuscript. This addition highlights the differences in apoptotic responses and further supports our conclusions regarding the selective effects of DCA and CU. Additionally, we have included the results of DS treatments to provide a more comprehensive analysis.
Comment 7. Comments on the Quality of English Language. There are some inaccuracies and syntactic errors, especially in the introduction section. To give a few examples, Line 56 : "relapse rates still need to be LOWER". Line 69: "TPI has not only GAINED significant attention".
Repply:
We appreciate your attention to detail and have carefully reviewed the introduction section for grammatical and syntactic errors.
In addition to these corrections, we have conducted a comprehensive review of the entire manuscript to identify and rectify any other inaccuracies or syntactic errors. We believe these revisions enhance the clarity and readability of our work.
Round 2
Reviewer 2 Report
Comments and Suggestions for Authors
After this round of revision, the authors notably improved the quality of the data presented. They now correctly underline the limitations of a study that focuses on a single model (a cell line), but the conclusions that they draw are solid within the chosen model.
As for describing the results on the combinatorial/ sequential administration of DS and CU, the fact that dTPI inactivation is primarily driven by the first compound they introduced may confirm that there is a competition for the same binding pocket (i.e. the first compound occupies the pocket and therefore the second one has less access).
Aside from these interpretations, since the authors fully answered to all the points I raised, I believe that now the quality of their work is suitable to publication in this journal.
Comments on the Quality of English LanguageI still noticed a few little mistakes, more conceptual than grammatical. For example, when in the introduction they state:
"While significant progress has been made in ALL treatment modalities, including chemotherapy, targeted therapies, and stem cell transplantation, relapse rates still need to be higher, and treatment resistance is a substantial challenge"
I think what they mean is: "although significant progress has been made in ALL treatments... relapse rates still remain high (especially for some groups of patients), and treatment resistance is a substantial challenge".
Author Response
Dear Reviewers,
We sincerely appreciate the time and effort you have dedicated to reviewing our manuscript titled “Deamidated triosephosphate isomerase as a selective target for T-ALL
therapy: Synergistic inhibition by dichloroacetic acid and curcumin.” (biomolecules-3202107). Your insightful comments and constructive feedback have been invaluable in improving the quality and clarity of our work.
We have carefully considered each of your comments and suggestions. Below, we provide detailed, point-by-point responses to all the issues raised. Changes made to the manuscript have been highlighted in red for your convenience.
Response to reviewer #2
Comment 1. Comments and Suggestions for Authors.
After this round of revision, the authors notably improved the quality of the data presented. They now correctly underline the limitations of a study that focuses on a single model (a cell line), but the conclusions that they draw are solid within the chosen model.
As for describing the results on the combinatorial/ sequential administration of DS and CU, the fact that dTPI inactivation is primarily driven by the first compound they introduced may confirm that there is a competition for the same binding pocket (i.e. the first compound occupies the pocket and therefore the second one has less access).
Aside from these interpretations, since the authors fully answered to all the points I raised, I believe that now the quality of their work is suitable to publication in this journal.
Repply:
We appreciate the reviewer’s thoughtful feedback and are grateful for the positive evaluation of our revised manuscript. We are pleased to hear that the improvements made to the data presentation and the discussion of the study's limitations were satisfactory.
Regarding the interpretation of the combinatorial or sequential administration of DS and DCA, we agree with the insightful suggestion that dTPI inactivation may be driven by competitive binding for the same pocket. This hypothesis aligns with our observations, and we will continue to explore this mechanism in future studies.
Thank you again for your valuable comments and the recommendation for publication.
Comment 2. Comments on the Quality of English Language.
I still noticed a few little mistakes, more conceptual than grammatical. For example, when in the introduction they state:
"While significant progress has been made in ALL treatment modalities, including chemotherapy, targeted therapies, and stem cell transplantation, relapse rates still need to be higher, and treatment resistance is a substantial challenge"
I think what they mean is: "although significant progress has been made in ALL treatments... relapse rates still remain high (especially for some groups of patients), and treatment resistance is a substantial challenge".
Repply:
We sincerely thank the reviewer for their detailed observations regarding the quality of the English language and conceptual clarity. We acknowledge the points raised, particularly the example from the introduction, and agree that the suggested wording more accurately conveys the intended message.
In response to this feedback, we have carefully reviewed and revised the entire manuscript, ensuring conceptual accuracy and linguistic precision throughout. The suggested improvement to the introduction has been incorporated as follows:
"Although significant progress has been made in ALL treatments, including chemotherapy, targeted therapies, and stem cell transplantation, relapse rates remain high, especially for certain patient groups, and treatment resistance continues to be a substantial challenge."
We believe this revision, along with the other improvements made across the text, addresses the reviewer's concerns and enhances the overall quality of the manuscript. We greatly appreciate the reviewer's valuable input in helping us improve the clarity and readability of our work.